# Estimation and imputation in Probabilistic Principal Component Analysis with Missing Not At Random data

**Aude Sportisse**
Sorbonne University
Paris, France
aude.sportisse@sorbonne-universite.fr

**Claire Boyer**
Sorbonne University
Paris, France
claire.boyer@sorbonne-universite.fr

**Julie Josse**
INRIA
Montpellier, France
julie.josse@inria.fr

## Abstract

Missing Not At Random (MNAR) values where the probability of having missing data may depend on the missing value itself, are notoriously difficult to account for in analyses, although very frequent in the data. One solution to handle MNAR data is to specify a model for the missing data mechanism, which makes inference or imputation tasks more complex. Furthermore, this implies a strong *a priori* on the parametric form of the distribution. However, some works have obtained guarantees on the estimation of parameters in the presence of MNAR data, without specifying the distribution of missing data [18, 25]. This is very useful in practice, but is limited to simple cases such as few self-masked MNAR variables in data generated according to linear regression models. We continue this line of research, but extend it to a more general MNAR mechanism, in a more general model of the probabilistic principal component analysis (PPCA), *i.e.*, a low-rank model with random effects. We prove identifiability of the PPCA parameters. We then propose an estimation of the loading coefficients, and a data imputation method. Both are based on estimators of means, variances and covariances of missing variables, for which consistency is discussed. These estimators have the great advantage of being calculated using only the observed information, leveraging the underlying low-rank structure of the data. We illustrate the relevance of the method with numerical experiments on synthetic data and also on two datasets, one collected from a medical register and the other one from a recommendation system.

## 1 Introduction

The problem of missing data is ubiquitous in the practice of data analysis. Theoretical guarantees of estimation strategies or imputation methods rely on assumptions regarding the missing-data mechanism, *i.e.* the cause of the lack of data. Rubin [22] introduced three missing-data mechanisms. The data are said (i) Missing Completely At Random (MCAR) if the probability of being missing does not depend on any values observed or missing, (ii) Missing At Random (MAR) if the probability of being missing only depends on observed values, (iii) Missing Not At Random (MNAR) if the unavailability of the data may depend on both observed and unobserved data such as its value itself. We focus on this later case, which is frequent in practice, and theoretically challenging. A classic example of MNAR data is surveys about salary for which rich people would be less willing to disclose their income.

When the data is MCAR or MAR, statistical inference is carried out by ignoring the missing data mechanism [10]. In the MNAR case, the observed data are no longer representative of the population, which leads to selection bias in the sample, and therefore to bias in the parameters estimation when using for instance complete case analysis. One solution to handle MNAR data, known as *selection model* [10], is to model missing data distribution; most of the time, by logistic regression models

[4, 19, 24]. This comes at the price of an important computational burden to perform inference and is often restricted to a limited number of MNAR variables. In the recommender system community, some authors [12, 3, 11, 28] suggest that not MCAR values can be handled using a joint modelling of the data and mechanism distributions by matrix factorization; then they debias existing methods for MCAR data, for instance with inverse probability weighting approaches.

In addition, a key issue of MNAR data is to establish identifiability, which is not always guaranteed [15]. The literature on this topic is abundant, both in the non-parametric [17, 16, 6, 23, 20], and semi-parametric settings [27, 14]. For parametric models, in the case of multivariate regression, Tang et al. [25] and Miao et al. [15] guarantee the identifiability of the coefficients of the conditional distribution of $Y|X$, when $Y$ is missing. Tang et al. [25] estimate them by calculating the coefficient of the distributions of $X$ and $X|Y$ using only observations with no missing values. Besides, in a linear model with self-masked missing mechanism, *i.e.*, the lack depends only on the missing variable itself, Mohan et al. [18] consider a related approach based on graphical models, adopting a causal point of view. Despite the great advantage of not modeling the distribution of missing values, the assumption of a self-masked MNAR mechanism and the restriction to a linear model are yet strong.

**Contributions.** We consider a framework where the data are generated according to a probabilistic principal components analysis (PPCA) [26] model. Contrary to available works that handle only MAR data in PPCA [5], we consider that the missing values mechanism can be MNAR (on several variables) and we also consider the possibility of having different mechanisms in the same data (MNAR and M(C)AR).

- We prove the identifiability of the PPCA model parameters in a self-masked MNAR values setting encompassing a large set of self-masked mechanism distributions.

- For more general MNAR mechanism, we give a strategy to estimate the PPCA loading parameters without any modeling of the missing-data mechanism and use it to impute missing values.

- The proposed method is based on estimators for the mean, the variance and the covariance of the variables with MNAR values. We show that they can be consistently estimated. Two strategies lead to the proposed estimators: (i) the first one uses algebraic arguments based on partial linear models derived from the PPCA model; (ii) the second one is inspired by [18] and uses graphical models and in particular the so-called missingness graph.

- We derive an algorithm implementing our proposal. We show that it outperforms the state-of-the-art methods on synthetic data and on two real datasets, collected from a medical registry (Traumabase®) and from a joke recommender system (the Jester Online Joke Recommender System [2]). The code to reproduce all the simulations and the numerical experiments is available at `https://github.com/AudeSportisse/PPCA_MNAR`.

## 2 PPCA model with informative missing values: identifiability issues

**Setting.** The data matrix $Y \in \mathbb{R}^{n \times p}$ is assumed to be generated under a fully-connected PPCA model [26] (a.k.a. a low-rank model with random effects), *i.e.* by the factorization of the loading matrix $B \in \mathbb{R}^{r \times p}$ and $r$ latent variables grouped in the matrix $W \in \mathbb{R}^{n \times r}$,

$$Y = \mathbf{1}\alpha + WB + \epsilon, \text{ with } \begin{cases} W = (W_{1.}|\dots|W_{n.})^T, \text{ with } W_{i.} \sim \mathcal{N}(0_r, \mathrm{Id}_{r \times r}) \in \mathbb{R}^r, \\ B \text{ of rank } r < \min\{n, p\}, \\ \alpha \in \mathbb{R}^p \text{ and } \mathbf{1} = (1 \dots 1)^T \in \mathbb{R}^n, \\ \epsilon = (\epsilon_{1.}|\dots|\epsilon_{n.})^T, \text{ with } \epsilon_{i.} \sim \mathcal{N}(0_p, \sigma^2 \mathrm{Id}_{p \times p}) \in \mathbb{R}^p, \end{cases} \tag{1}$$

for $\sigma^2$ and $r$ known. In the sequel, $Y_{.j}$ and $Y_{i.}$ respectively denote the column $j$ and the row $i$ of $Y$. The rows of $Y$ are identically distributed, $\forall i \in \{1, \dots, n\}$, $Y_{i.} \sim \mathcal{N}(\alpha, \Sigma)$, with $\Sigma = B^T B + \sigma^2 \mathrm{Id}_{p \times p}$. We denote $\Omega \in \{0, 1\}^{n \times p}$ the missing-data pattern (or mask) defined as follows:

$$\forall i \in \{1, \dots, n\}, \forall j \in \{1, \dots, p\}, \quad \Omega_{ij} = \begin{cases} 0 & \text{if } Y_{ij} \text{ is missing}, \\ 1 & \text{otherwise}. \end{cases} \tag{2}$$

Some variables $Y_{.m_1}, \dots, Y_{.m_d}$, indexed by $\mathcal{M} := \{m_1, \dots, m_d\} \subset \{1, \dots, p\}$ (with $d < p$), contain MNAR values. The other variables are considered to be observed (or M(C)AR see Appendix B.5). We define a general MNAR mechanism where the probability to have missing values may

depend on the $d$ MNAR variables but also on $p - d - r$ other variables that can be observed or M(C)AR[1]. The remaining $r$ variables are called pivot variables and can be observed or MCAR. More precisely, we denote the complementary of a set $\mathcal{A}$ as $\overline{\mathcal{A}} := \{1, \ldots, p\} \setminus \mathcal{A}$. The general MNAR mechanism is defined as follows, with $\mathcal{J} \subset \overline{\mathcal{M}}$ the set of indices of the $r$ pivot variables ($|\mathcal{J}| = r$),

$$\forall m \in \mathcal{M}, \forall i \in \{1, \ldots, n\}, \quad \mathbb{P}(\Omega_{im} = 1 | Y_{i.}) = \mathbb{P}(\Omega_{im} = 1 | (Y_{ik})_{k \in \overline{\mathcal{J}}}). \qquad (3)$$

We also define a specific MNAR mechanism, called the self-masked MNAR mechanism as follows. We assume that $d$ variables are self-masked MNAR indexed by $\mathcal{M}$ and the $p - d$ other variables are MCAR (or observed), indexed by $\overline{\mathcal{M}}$, i.e, $\forall i \in \{1, \ldots, n\}$,

$$\forall m \in \mathcal{M}, \quad \mathbb{P}(\Omega_{im} = 1 | Y_{i.}) = \mathbb{P}(\Omega_{im} = 1 | Y_{im}). \qquad (4)$$

**Model identifiability.** We prove the identifiability of the PPCA model (see Appendix A for the complete proof), *i.e.* the joint distribution of $Y$ can be uniquely determined from the available information, in the self-masked missing values case. More particularly, assume the following

**A01.** $d$ variables are self-masked MNAR as in (4) and the $p - d$ other variables are MCAR (or observed). The missing-data distributions $(F_m)_{m \in \mathcal{M}}$ and $(F_j)_{j \in \overline{\mathcal{M}}}$ are known strictly monotone functions with a finite support, defined as follows, $\forall i \in \{1, \ldots, n\}$,

$$\forall m \in \mathcal{M}, \quad \mathbb{P}(\Omega_{im} = 1 | Y_{i.}) = F_m(\phi_m^0 + \phi_m^1 Y_{im}),$$
$$\forall j \in \overline{\mathcal{M}}, \quad \mathbb{P}(\Omega_{ij} = 1 | Y_{i.}) = \mathbb{P}(\Omega_{ij} = 1) = F_j(\phi_j),$$

with $\phi_j \in \mathbb{R}$ and $\phi_m = (\phi_m^0, \phi_m^1) \in \mathbb{R}^2$ the mechanism parameters.

**A02.** $\forall (k, \ell) \in \{1, \ldots, p\}^2, \quad k \neq \ell, \quad \Omega_{.k} \perp\!\!\!\perp \Omega_{.\ell} | Y$

Note that under Assumption **A01.**, any function $F_m, m \in \mathcal{M}$ can be considered, as a logistic function while [15] presented many counterexamples when identification fails considering the logistic distribution. **A02.** requires that the missing-data patterns are independent conditionally to the data.

**Proposition 1.** *Under Assumptions **A01.** and **A02.**, the parameters $(\alpha, \Sigma)$ of the PPCA model* (1) *and the mechanism parameters $\phi = (\phi_\ell)_{\ell \in \{1, \ldots p\}}$ are identifiable. Assuming that the noise level $\sigma^2$ is known, the parameter $B$ is identifiable up to a row permutation.*

## 3 Estimators with theoretical guarantees

In this section, we provide estimators of the means, variances and covariances for the MNAR variables, when data are generated under the PPCA model described in (1). These estimators are used to derive an estimator of the loading matrix $B$ in (1). This makes it possible to derive a new imputation method with MNAR data as detailed in Algorithm 1.

We denote $\mathcal{J}_{-j} := \mathcal{J} \setminus \{j\}$ and assume

**A1.** $\forall m \in \mathcal{M}, \forall j \in \mathcal{J}, \; \left( B_{.m} \quad (B_{.j'})_{j' \in \mathcal{J}_{-j}} \right)$ is invertible,

**A2.** $\forall m \in \mathcal{M}, \forall j \in \mathcal{J}, \; Y_{.j} \perp\!\!\!\perp \Omega_{.m} | (Y_{.k})_{k \in \overline{\{j\}}}.$

Note that Assumption **A1.** implies that $B$ has a full rank $r$ and that any variable in $Y$ is generated by all the latent variables[2] (named a "fully-connected" PPCA). Assumption **A2.** is implied by the general MNAR mechanism in (3).

We start by illustrating the methodology and the assumptions using an example in small dimension, before turning to the general case.

### 3.1 Estimation of the mean of a MNAR variable

Consider a toy dataset where $p = 3, r = 2$, in which only one variable is missing, $\mathcal{M} = \{1\}$ and there are two pivots variables $\mathcal{J} = \{2, 3\}$. Note that the MNAR mechanism is self-masked in such a context, because Equation (3) leads to $\mathbb{P}(\Omega_{.1} = 1 | Y_{.1}, Y_{.2}, Y_{.3}) = \mathbb{P}(\Omega_{.1} = 1 | Y_{.1})$, but the method can be extended to more general cases. Our aim is to estimate the mean of $Y_{.1}$, without specifying the distribution of the missing-data mechanism.

**Using algebraic arguments.** We proceed in three steps: (i) **A1.** allows to obtain linear link between the pivot variables $(Y_{.2}, Y_{.3})$ and the MNAR variable $Y_{.1}$. For instance,

$$Y_{.2} = \mathcal{B}_{2\to1,3[0]} + \mathcal{B}_{2\to1,3[1]}Y_{.1} + \mathcal{B}_{2\to1,3[3]}Y_{.3} + \zeta, \tag{5}$$

with $\zeta$ a noise term, $\mathcal{B}_{2\to1,3[0]}$, $\mathcal{B}_{2\to1,3[1]}$ and $\mathcal{B}_{2\to1,3[3]}$ the intercept and the coefficients in the model (the arrow $2 \to 1, 3$ indicates the regression model of $Y_{.2}$ on $Y_{.1}$ and $Y_{.3}$, while the squared bracket represents the coefficient, for instance 3 for the coefficient of $Y_{.3}$) ; (ii) Assumption **A2.**, *i.e.* $Y_{.2} \perp\!\!\!\perp \Omega_{.1}|Y_{.1}, Y_{.3}$, is required to obtain identifiable and consistent parameters of the distribution of $Y_{.2}$ given $Y_{.1}, Y_{.3}$ in the complete-case when $\Omega_{.1} = 1$, denoted as $\mathcal{B}^c_{2\to1,3[0]}$, $\mathcal{B}^c_{2\to1,3[1]}$ and $\mathcal{B}^c_{2\to1,3[3]}$,

$$(Y_{.2})_{|\Omega_{.1}=1} = \mathcal{B}^c_{2\to1,3[0]} + \mathcal{B}^c_{2\to1,3[1]}Y_{.1} + \mathcal{B}^c_{2\to1,3[3]}Y_{.3} + \zeta^c, \tag{6}$$

(note that the regression of $Y_{.1}$ on $(Y_{.2}, Y_{.3})$ is prohibited, as **A2.** does not hold); (iii) using again **A2.**,

$$\mathbb{E}\left[Y_{.2}|Y_{.1}, Y_{.3}, \Omega_{.1} = 1\right] = \mathbb{E}\left[\mathcal{B}^c_{2\to1,3[0]} + \mathcal{B}^c_{2\to1,3[1]}Y_{.1} + \mathcal{B}^c_{2\to1,3[3]}Y_{.3}|Y_{.1}, Y_{.3}\right],$$

and taking the expectation leads to

$$\mathbb{E}\left[Y_{.2}\right] = \mathcal{B}^c_{2\to1,3[0]} + \mathcal{B}^c_{2\to1,3[1]}\mathbb{E}\left[Y_{.1}\right] + \mathcal{B}^c_{2\to1,3[3]}\mathbb{E}\left[Y_{.3}\right].$$

The latter expression can be reshuffled so that the expectation of $Y_{.1}$ can be estimated: the means of $Y_{.2}$ and $Y_{.3}$ are estimated by standard empirical estimators (it will be Assumption **A4.** in the sequel).

**Using graphical arguments.** The PPCA model can be represented with structural causal graphs [21], as illustrated in Figure 1. The top left graph in which each variable is generated by a combination of all latent variables, see Assumption **A1.**, can be represented as the top right one, as $Y_{.1} \leftarrow W_{.1} \to Y_{.2}$ is equivalent to $Y_{.1} \leftrightarrow Y_{.2}$ (see [21, page 52]). Then, six reduced graphical models can be derived from the top right graph (two instances are represented in the bottom). Indeed, a bidirected edge $Y_{.1} \leftrightarrow Y_{.2}$ can be interchanged (see [21, rule 1, page 147]) with an oriented edge $Y_{.1} \to Y_{.2}$, if each neighbor of $Y_{.2}$ (*i.e.* $Y_{.1}$ or $Y_{.3}$) is inseparable of $Y_{.1}$ (see [21, page 17]). The bottom left graph can also be represented by Equation (6), which gives a connection between the algebraic and graphical approaches.

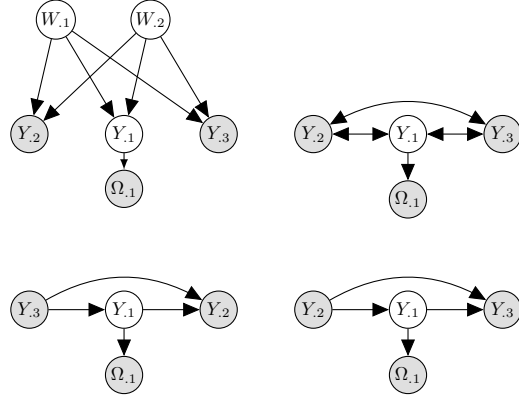

Figure 1: Graphical models for the toy example with one missing variable $Y_{.1}$, $p = 3$ and $r = 2$.

## 3.2 Estimation of the mean, variance and covariances of the MNAR variables

In a general case, estimators of the mean, variance and covariances of the variables with MNAR values can be computed one by one. We detail the results only for one variable $Y_{.m}, m \in \mathcal{M}$, but the results hold for several variables with MNAR values. In addition, the other variables are considered to be observed for simplicity but they could contain MCAR and MAR values as well, as explained in Appendix B.5. We adopt the algebraic strategy here to derive estimators (see Appendix B for proofs) but graphical arguments can also be used to obtain similar results (see Appendix F). The starting point is to exploit the linear links between variables, as described in the next lemma.

**Lemma 2.** *Under the PPCA model* (1) *and Assumption **A1.**, choose $j \in \mathcal{J}$. One has*

$$Y_{.j} = \mathcal{B}_{j\to m,\mathcal{J}_{-j}[0]} + \sum_{j'\in\mathcal{J}_{-j}} \mathcal{B}_{j\to m,\mathcal{J}_{-j}[j']}Y_{.j'} + \mathcal{B}_{j\to m,\mathcal{J}_{-j}[m]}Y_{.m} + \zeta, \tag{7}$$

*where $\zeta = -\sum_{j'\in\mathcal{J}_{-j}} \mathcal{B}_{j\to m,\mathcal{J}_{-j}[j']}\epsilon_{.j'} - \mathcal{B}_{j\to m,\mathcal{J}_{-j}[m]}\epsilon_{.m} + \epsilon_{.j}$. is a noise term.*

*$\mathcal{B}_{j\to m,\mathcal{J}_{-j}[0]}$, $\mathcal{B}_{j\to m,\mathcal{J}_{-j}[j']}$ and $\mathcal{B}_{j\to m,\mathcal{J}_{-j}[m]}$ are given in Appendix B.1 and depend on the coefficients of $B$ given in* (1).

Then we define the regression coefficients of $Y_{.j}$ on $Y_{.m}$ and $Y_{.k}$, for $k \in \mathcal{J}_{-j}$ in the complete case, that will be used to express the mean of a variable with MNAR values.

**Definition 3** (Coefficients in the complete case). *For $j \in \mathcal{J}$ and $k \in \mathcal{J}_{-j}$, let $\mathcal{B}^c_{j \to m, \mathcal{J}_{-j}[0]}$, $\mathcal{B}^c_{j \to m, \mathcal{J}_{-j}[m]}$ and $\mathcal{B}^c_{j \to m, \mathcal{J}_{-j}[j']}$ be respectively the intercept and the coefficients standing for the effects of $Y_{.j}$ on $(Y_{.m}, (Y_{.j'})_{j' \in \mathcal{J}_{-j}})$ in the complete case, i.e. when $\Omega_{.m} = 1$:*

$$(Y_{.j})_{|\Omega_{.m}=1} := \mathcal{B}^c_{j \to m, \mathcal{J}_{-j}[0]} + \sum_{j' \in \mathcal{J}_{-j}} \mathcal{B}^c_{j \to m, \mathcal{J}_{-j}[j']} Y_{.j'} + \mathcal{B}^c_{j \to m, \mathcal{J}_{-j}[m]} Y_{.m} + \zeta^c, \qquad (8)$$

*with $\zeta^c = -\sum_{j' \in \mathcal{J}_{-j}} \mathcal{B}^c_{j \to m, \mathcal{J}_{-j}[j']} \epsilon_{.j'} - \mathcal{B}^c_{j \to m, \mathcal{J}_{-j}[m]} \epsilon_{.m} + \epsilon_{.j}$.*

Then, we make the two following assumptions:

**A3.** For all $j \in \mathcal{J}$, for all $m \in \mathcal{M}$, the complete-case coefficients $\mathcal{B}^c_{j \to m, \mathcal{J}_{-j}[0]}$, $\mathcal{B}^c_{j \to m, \mathcal{J}_{-j}[m]}$ and $\mathcal{B}^c_{j \to m, \mathcal{J}_{-j}[k]}$, $k \in \mathcal{J}_{-j}$ can be consistently estimated.

**A4.** The means $(\alpha_j)_{j \in \mathcal{J}}$, variances $(\mathrm{Var}(Y_{.j}))_{j \in \mathcal{J}}$ and covariances $(\mathrm{Cov}(Y_{.j}, Y_{.j'}))_{j \in \mathcal{J}, j' \in \mathcal{J}_{-j}}$ of the $r$ pivot variables can be consistently estimated.

Note that Assumption **A4.** is met whether the $r$ pivot variables are fully observed.

**Proposition 4** (Mean estimator). *Consider the PPCA model* (1). *Under Assumptions A1. and A2., an estimator of the mean of a MNAR variable $Y_{.m}$, for $m \in \mathcal{M}$, can be constructed as follows: choose $j \in \mathcal{J}$, and compute*

$$\hat{\alpha}_m := \frac{\hat{\alpha}_j - \hat{\mathcal{B}}^c_{j \to m, \mathcal{J}_{-j}[0]} - \sum_{j' \in \mathcal{J}_{-j}} \hat{\mathcal{B}}^c_{j \to m, \mathcal{J}_{-j}[j']} \hat{\alpha}_{j'}}{\hat{\mathcal{B}}^c_{j \to m, \mathcal{J}_{-j}[m]}}, \qquad (9)$$

*with $(\hat{\mathcal{B}}^c_{j \to m, \mathcal{J}_{-j}[k]})_{k \in \{0,m\} \cup \mathcal{J}_{-j}}$ estimators of the coefficients obtained from Definition 3.*

*Under the additional Assumptions A3. and A4., this estimator is consistent.*

The proof is given in Appendix B.2. Proposition 4 provides an estimator easily computable from all observed cells. Furthermore, different choices of $Y_{.j}$, $j \in \mathcal{J}$ can be done in Equation (9) and all the resulting estimators may be aggregated to stabilize the estimation of $\alpha_m$.

**Proposition 5** (Variance and covariances estimators). *Consider the PPCA model* (1). *Under Assumptions A1. and A2., an estimator of the variance of a MNAR variable $Y_{.m}$, for $m \in \mathcal{M}$, and its covariances with the pivot variables, can be constructed as follows: choose a pivot variable $Y_{.j}$ for $j \in \mathcal{J}$ and compute*

$$\left( \widehat{\mathrm{Var}}(Y_{.m}) \quad \widehat{\mathrm{Cov}}(Y_{.m}, (Y_{.j'})_{j' \in \mathcal{J}}) \right)^T := (\widehat{M_j})^{-1} \widehat{P}_j, \qquad (10)$$

*assuming that $\sigma^2$ tends to zero, with $\widehat{M_j}^{-1} \in \mathbb{R}^{(r+1) \times (r+1)}$, $\widehat{P}_j \in \mathbb{R}^{r+1}$ detailed in Appendix B.3. These quantities depend on $(\hat{\alpha}_{j'})_{j' \in \mathcal{J}}$, $\hat{\alpha}_m$ given in Proposition 4, on $(\widehat{\mathrm{Var}}(Y_{.j'}))_{j' \in \mathcal{J}}$ and on complete-case coefficients such as $(\hat{\mathcal{B}}^c_{j' \to m, \mathcal{J}_{-j'}[k]})_{k \in \{m\} \cup \mathcal{J}_{-j'}}$ for $j' \in \mathcal{J}$.*

*Under the additional Assumptions A3. and A4., the estimators of the variance of $Y_{.m}$ and its covariances with the pivot variables given in* (10) *are consistent.*

The proof is given in Appendix B.3. Note that to estimate the variance of a MNAR variable, only $r$ pivot variables are required to solve (10) and $r$ tasks have to be performed for estimating the coefficients of the effects of $Y_{.k}$ on $(Y_{.\ell})_{\ell \in \{m\} \cup \mathcal{J}_{-k}}$ for all $k \in \mathcal{J}$.

All the ingredients can be combined to form an estimator $\hat{\Sigma}$ for the covariance matrix $\Sigma$. Define

$$\hat{\Sigma} := \left( \widehat{\mathrm{Cov}}(Y_{.k}, Y_{.\ell}) \right)_{k, \ell \in \{1, \dots, p\}}, \qquad (11)$$

- if $Y_{.k}$ and $Y_{.\ell}$ have both consistent mean/variance estimators, then $\widehat{\mathrm{Cov}}(Y_{.k}, Y_{.\ell})$ can be trivially evaluated by standard empirical covariance estimators.

- if $Y_{.k}$ is a MNAR variable and $Y_{.\ell}$ is a pivot variable, then $\widehat{\mathrm{Cov}}(Y_{.k}, Y_{.\ell})$ is given by (10),

- if $Y_{.k}$ is a MNAR variable and $Y_{.\ell}$ is not a pivot variable, *i.e.* $\ell \in \overline{\mathcal{J}} \setminus \{k\}$, a similar strategy as the one above can be devised. Then $\widehat{\mathrm{Cov}}(Y_{.k}, Y_{.\ell})$ is given by (48) detailed in Appendix B.4 and for which some additional assumptions similar as the ones above are required. This estimator relies on the choice of $r - 1$ pivot variables indexed by $j$ and $\mathcal{H} \subset \mathcal{J}$, and only necessitates to evaluate the effects of $Y_{.j}$ on $(Y_{.j'})_{j' \in \{k,\ell\} \cup \mathcal{H}}$ in the complete case.

## 3.3 Performing PPCA with MNAR variables

With the estimator $\hat{\Sigma}$ in (11) at hand, one can perform the estimation of the loading matrix $B$ in (1).

**Definition 6** (Estimation of the loading matrix). *Given the estimator $\hat{\Sigma}$ of the covariance matrix in (11), let the orthogonal matrix $\hat{U} = (\hat{u}_1| \ldots |\hat{u}_p) \in \mathbb{R}^{p \times p}$ and the diagonal matrix $\hat{D} = \mathrm{diag}(\hat{d}_1, \hat{d}_2, \ldots, \hat{d}_p) \in \mathbb{R}^{p \times p}$ with $\hat{d}_1 \geq \hat{d}_2 \geq \ldots \geq \hat{d}_p \geq 0$ form the singular value decomposition of the following matrix $\hat{\Sigma} - \sigma^2 \mathrm{Id}_{p \times p} =: \hat{U} \hat{D} \hat{U}^T$. An estimator $\hat{B}$ of $B$ can be defined using the $r$ first singular values and vectors, as follows*

$$\hat{B} = \hat{D}_{|r}^{1/2} \hat{U}_{|r}^T = \mathrm{diag}(\hat{d}_1, \ldots, \hat{d}_r)^{1/2} (\hat{u}_1^T| \ldots |\hat{u}_r^T)^T \tag{12}$$

The estimation of the loading matrix is used to impute the variables with missing values. More precisely, a classical strategy to impute missing values is to estimate their conditional expectation given the observed values. One can note that with $\Sigma = B^T B + \sigma^2 \mathrm{Id}_{p \times p}$, the conditional expectation of $Y_{.m}$ for $m \in \mathcal{M}$ given $(Y_{.k})_{k \in \overline{\mathcal{M}}}$ reads as follows

$$\mathbb{E}[Y_{.m}|(Y_{.k})_{k \in \overline{\mathcal{M}}}] = \alpha_m + \Sigma_{m,\overline{\mathcal{M}}} \Sigma_{\overline{\mathcal{M}},\overline{\mathcal{M}}}^{-1} \left( Y_{.\overline{\mathcal{M}}}^T - \alpha_{\overline{\mathcal{M}}} \right),$$

with $\Sigma_{m,\overline{\mathcal{M}}} := (\Sigma_{m,k})_{k \in \overline{\mathcal{M}}}^T$, $\Sigma_{\overline{\mathcal{M}},\overline{\mathcal{M}}} := (\Sigma_{k,k'})_{k,k' \in \overline{\mathcal{M}}}$, $Y_{.\overline{\mathcal{M}}} := (Y_{.k})_{k \in \overline{\mathcal{M}}}$, and $\alpha_{\overline{\mathcal{M}}} := (\alpha_k)_{k \in \overline{\mathcal{M}}}$.

**Definition 7** (Imputation of a MNAR variable). *Set $\hat{\Gamma} := \hat{B}^T \hat{B} + \sigma^2 \mathrm{Id}_{p \times p}$ for $\hat{B}$ given in Definition 6. The MNAR variable $Y_{.m}$ with $m \in \mathcal{M}$ can be imputed as follows: for $i$ such that $\Omega_{i,m} = 0$,*

$$\hat{Y}_{im} = \hat{\alpha}_m + \hat{\Gamma}_{m,\overline{\mathcal{M}}} \hat{\Gamma}_{\overline{\mathcal{M}},\overline{\mathcal{M}}}^{-1} \left( Y_{i,\overline{\mathcal{M}}}^T - \hat{\alpha}_{\overline{\mathcal{M}}} \right) \tag{13}$$

*with $\hat{\Gamma}_{m,\overline{\mathcal{M}}} := (\hat{\Gamma}_{m,k})_{k \in \overline{\mathcal{M}}}^T$, $\hat{\Gamma}_{\overline{\mathcal{M}},\overline{\mathcal{M}}} := (\hat{\Gamma}_{k,k'})_{k,k' \in \overline{\mathcal{M}}}$, $Y_{.\overline{\mathcal{M}}} := (Y_{.k})_{k \in \overline{\mathcal{M}}}$ and $\hat{\alpha}_{\overline{\mathcal{M}}} := (\hat{\alpha}_k)_{k \in \overline{\mathcal{M}}}$.*

## 3.4 Algorithm

The proposed imputation method described in Algorithm 1 can handle the different MNAR mechanisms, the self-masked MNAR case and the general MNAR cases where the probability to have missing values on variables depends on both the underlying values and values of other variables (observed or missing).

---

**Algorithm 1** PPCA with MNAR variables.

---

**Require:** $r$ (number of latent variables), $\sigma^2$ (noise level), $\mathcal{J}$ (pivot variables indices), $\Omega$ (mask).

1: **for** each MNAR variable $(Y_{.m})_{m \in \mathcal{M}}$ **do**
2:      Evaluate $\hat{\alpha}_m$ the estimator of its mean given in (9) using the $r$ pivot variables indexed by $\mathcal{J}$.
3:      Evaluate $\widehat{\mathrm{Var}}(Y_{.m})$, and $\widehat{\mathrm{Cov}}(Y_{.m}, Y_{.\ell})$ for $\ell \in \mathcal{J}$, using (10).
4:      Evaluate $\widehat{\mathrm{Cov}}(Y_{.m}, Y_{.\ell})$ for $\ell \in \overline{\mathcal{J}} \setminus \{m\}$ using Proposition 8.
5: **end for**
6: Form $\hat{\Sigma}$, covariance matrix estimator in (11).
7: Compute the loading matrix estimator $\hat{B}$ given in (12).
8: Compute $\hat{\Gamma} = \hat{B}^T \hat{B} + \sigma^2 \mathrm{Id}_{p \times p}$.
9: **for** each missing variable $(Y_{.j})$ **do**
10:      **for** $i$ such that $\Omega_{ij} = 0$ **do**
11:          $\hat{Y}_{ij} \leftarrow$ Impute $Y_{ij}$ as in (13).
12:      **end for**
13: **end for**

---

Algorithm 1 requires the set $\mathcal{J}$, *i.e.* the selection of $r$ pivot variables on which the regressions in Propositions 4, 5 and 8 will be performed. If there are more than $r$ variables that can be pivot, we suggest selecting a bigger set $(> r)$ and computing the final estimator with the median of the estimators over all possible combinations. The efficiency of this strategy is illustrated in Appendix C.

The estimators associated to any missing variable in the steps 1 to 5 are computed in the complete case, i.e. with the rows for which the missing variable is observed. When the pivot variables are also missing, the complete case corresponds to discarding all rows where the pivot variables or the MNAR one are missing and not all rows containing missing values. This could be problematic in the high-dimensional setting, but here the low-rank assumption ($r < \min\{n, p\}$) ensures that the number of pivot variables is small enough, so that the complete case analysis will not result in discarding many rows of the dataset.

In order to estimate the coefficients in Definition 3, we use ordinary least squares despite that the exogeneity assumption, *i.e.* the noise term is independent of the covariates, does not hold. It still leads to accurate estimation in numerical experiments as shown in Section 4. Actually, the consistency required by Assumption **A3.** holds as the variance of the noise tends to 0.

# 4 Numerical experiments

## 4.1 Synthetic data

We empirically compare Algorithm 1 (**MNAR**) to the state-of-the-art methods, including

- (i) **MAR**: our method which has been adapted to handle MAR data (inspired by [18, Theorems 1, 2, 3] in linear models), see Appendix G for details;
- (ii) **EMMAR**: which consists in an EM algorithm to perform PPCA with MAR values [5];
- (iii) **SoftMAR**: a matrix completion method using an iterative soft-thresholding singular value decomposition algorithm [13] relevant only for M(C)AR values;
- (iv) **MNARparam**: a matrix completion technique modeling the MNAR mechanism with a parametric logistic model [24].

Note that Method (ii) is specially designed to estimate the PPCA loading matrix and not to perform imputation, but this is possible combining Method (ii) with steps 8 and 9 in Algorithm 1. This is the other way around for completion Methods (iii) and (iv), but the loading matrix can be computed as in (12). Note also that Methods (iii) and (iv) are developed in a context of low-rank models with fixed effects. They require tuning a regularization parameter $\lambda$: we consider an oracle value minimizing the true imputation error. We also use oracle values for the noise level and the rank in Algorithm 1. These methods are compared with the imputation by the mean (**Mean**), which serves as a benchmark, and the naive listwise deletion method (**Del**) which consists in estimating the parameters empirically with the fully-observed data only. A comparison of the methods in terms of computational times is given in Appendix D.

**Measuring the performance.** For the loading matrix, the RV coefficient [8], which is a measure of relationship between two random vectors, is computed between the estimate $\hat{B}$ and the true $B$. An RV coefficient close to one means high correlation between the image spaces of $\hat{B}$ and $B$. Denoting the Frobenius norm as $\|.\|_F$, the quality of imputation is measured with the normalized imputation error given by $\|(\hat{Y} - Y) \odot (1 - \Omega)\|_F^2 / \|Y \odot (1 - \Omega)\|_F^2$.

**Setting.** We generate a data matrix of size $n = 1000$ and $p = 10$ from a PPCA model (1) with two latent variables ($r = 2$) and with a noise level $\sigma = 0.1$. Missing values are introduced on seven variables $(Y_{.k})_{k \in [1:7]}$ according to a logistic self-masked MNAR mechanism, leading to 35% of missing values in total. Results are presented[3] for one missing variable $Y_{.1}$ (same results hold for other missing variables). All the observed variables $(Y_{.k})_{k \in [8:10]}$ are considered to be pivot.

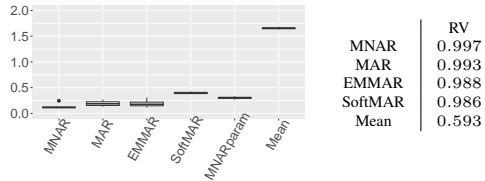

| | RV |
|---|---|
| MNAR | 0.997 |
| MAR | 0.993 |
| EMMAR | 0.988 |
| SoftMAR | 0.986 |
| Mean | 0.593 |

Figure 2: Imputation error (left) and median of the RV coefficients for the loading matrix (right).

Figure 3 shows that Algorithms 1 is the only one which always gives unbiased estimators of the mean, variance and associated covariances of $Y_{.1}$. As expected, the listwise deletion method provides biased estimates inasmuch as the observed sample is not representative of the population with MNAR data. Method (ii), specifically designed for PPCA models but assuming MAR missing values, provides biased estimators. Method (iv) improves on the benchmark mean imputation and on Method (iii) as well as it explicitly takes into account the MNAR mechanism, but it still leads to biased estimates probably because of the fixed effects model assumption. Figure 2 shows that Algorithm 1 gives the best estimate of the loading matrix and the smallest imputation error. Method (i), based on the same arguments as Algorithm 1 but considering MAR data, may be considered as a second choice for this low-dimensional example as the biais is quite small (yet not in higher dimension, see Appendix C).

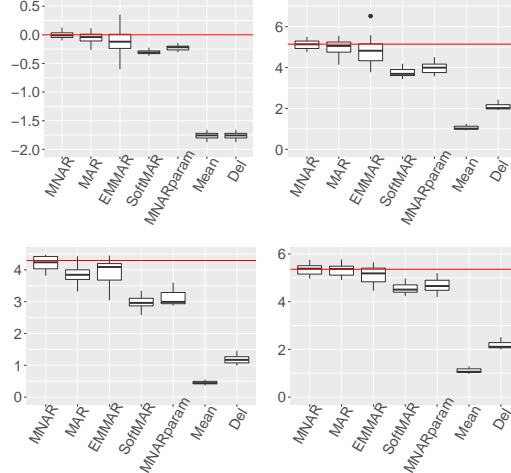

Figure 3: Mean (top left) and variance (top right) estimations of the missing variable and covariances (bottom) estimations of $\mathrm{Cov}(Y_{.1}, Y_{.2})$ (*i.e.* covariance between two missing variables) and of $\mathrm{Cov}(Y_{.1}, Y_{.8})$ (*i.e.* between one missing variable and one pivot variable). True values are indicated by red lines.

**Misspecification to the PPCA model.** The data matrix $Y \in \mathbb{R}^{n \times p}$ of size $n = 200$ and $p = 10$ is now generated under the fixed effects model such that $Y = \Theta + \epsilon$, with $\Theta \in \mathbb{R}^{n \times p}$ a low-rank matrix with $r = 2$ and $\epsilon \in \mathbb{R}^{n \times p}$ a Gaussian noise matrix with $\sigma = 0.1$. Figure 4 shows that mean and variance estimators given by Algorithm 1 have a larger variance than those given by Method (iv) precisely dedicated to this specific setting. But surprisingly, Algorithm 1 provides less biased estimates than Method (iv).

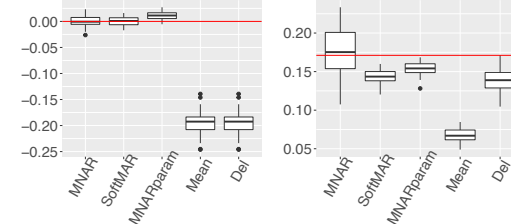

Figure 4: Mean (left) and variance (right) estimations of $Y_{.1}$ when data are generated under the fixed effects model.

In Appendix C, we report further simulation results, where we vary the features dimension ($p = 50$), the rank ($r = 5$), the missing values mechanism using probit self-masking and also multivariate MNAR (when the probability to be missing for a variable depends on its underlying values and on values of other variables that can be missing) and the percentage of missing values (10%, 50%). We obtain similar results as before, and as expected, all the methods deteriorate with an increasing percentage of missing values but our method remains stable.

In addition to the model misspecification experiment (assuming a fixed effect model), we assess the robustness of the methods in terms of noise level and we evaluate the impact of under- or overestimating the number $r$ of latent variables. When the level of noise increases, our method is very robust in terms of mean and variance estimations, and despite a bias for some covariances estimations for large noise it outperforms competitors regarding the imputation error. It also turns out that the procedure remains stable at a wrong specification of the number $r$ of latent variables.

## 4.2 Application to recommendation system data

To show the extent and feasibility of our methodology on real data, we detail the methodology on the Jester dataset [2] of 5000 users who rated 100 jokes, with 27% of missing values.

**Discussion on the assumptions.** First, considering MNAR and self-masking values is plausible because users only rate jokes they like or dislike strongly or might be ashamed to assume their taste for sexual jokes for instance. Then, Assumption **A1.**, which can be viewed as a low-rank assumption for the loading matrix, makes sense in the rating context: any variable (i.e. user preferences) can be

expressed as a linear combination of $r$ latent variables. In particular, the first latent variable opposes individuals who like jokes about physics but dislike jokes about sexuality, and conversely. Finally, Assumption **A2.** means that a user's non-response for a sexual joke given all jokes may depend on the scores of the sexual and physical jokes but not on the musical and computer jokes.

**Selecting the number $r$ of latent variables and estimating the noise variance.** In practice, to select $r$, one could use complete observations only but this is not possible when the number of features is large. As an alternative, we use a cross-validation strategy assuming M(C)AR mechanism as detailed in [7]. Algorithm 1 is robust to a misspecification of the rank (see Appendix C) and thus a reasonable heuristic may already be enough. With $r$ at hand, the noise variance is obtained directly using weighted residual sum of squares as in [9]. Without further information on the missing mechanisms, we select the $r$ pivot variables with the lowest missing rate.

**Imputation performances.** To assess the quality of our method, we introduce additional MNAR values using a logistic self-masked mechanism in a chosen variable with an initial rate of 33% and a final one of 65%. The other variables are considered M(C)AR. The process is repeated 10 times. We compare our method to the EMMAR, SoftMAR and add an imputation method based on deep generative models `Deep` [1][4]. The parametric method `MNARparam` is not performed as it does not scale on such large data. Figure 5 shows that Algorithm 1 outperforms the competitors (mean imputation corresponds to an error of 1).

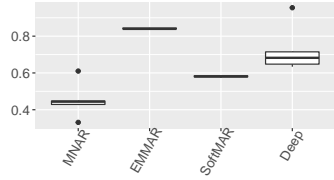

Figure 5: Imputation error for the Jester dataset.

### 4.3 Application to clinical data

We illustrate our method on the TraumaBase® dataset containing the clinical measurements of 3159 patients with brain trauma injury (see Appendix E for more information). Nine quantitative variables, selected by doctors, contain missing values (11% in the whole dataset). After discussion with doctors, some variables can be considered to have MNAR values, such as the variable *HR.ph*, which denotes the heart rate. Indeed, when the patient's condition is too critical and therefore his heart rate is either high or low, the heart rate may not be measured, as doctors prefer to provide emergency care.

As for the Jester dataset, we introduce additional MNAR values in the variable *HR.ph* (which has an initial missing rate of 1%) using a logistic self-masked mechanism leading to 50% missing values. Both the rank and the noise level are estimated using the complete-case analysis (1862 observations). The selection of the pivot variables was discussed with experts (doctors) who identified M(C)AR variables. In Figure 6, Algorithm 1 gives significantly smaller imputation error than other methods. In addition, a supervised learning task is also performed in Appendix E for which Algorithm 1 also gives the smallest prediction error.

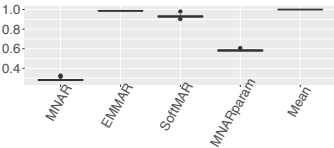

Figure 6: Imputation error for the TraumaBase dataset.

## Conclusion

In this work, we propose a new estimation and imputation method to perform PPCA with MNAR data (possibly coupled with M(C)AR data), without any need of modeling the missing mechanism. This comes with strong theoretical guarantees as identifiability and consistency, but also with an efficient algorithm. Estimating the rank in the PPCA setting with MNAR data remains non trivial. Once the number of latent variables is estimated, the noise variance can be estimated. A cross-validation strategy by additionally adding some MNAR values is a first solution, but this definitely requires further research. Another ambitious prospect would be to extend work to the exponential family to process count data, for example, which is prevalent in many application fields such as genomics.

## Broader impact

Our goal is to provide a rigorous and consistent method for processing MNAR missing values, in data with an underlying low-rank structure. The low-rank assumption has become widespread in applications in recent years and it plays a key modeling role in many scientific and engineering tasks, such as collaborative filtering, genome-wide studies, or even functional magnetic resonance imaging.

The problem of missing data is particularly evident for large data, possibly aggregated from multiple sources, that is why we illustrate this work on a real dataset such as the medical register TraumaBase, coming from different hospitals.

Managing informative missing data is a double challenge: on the one hand, because most of the available data contains missing values, preventing the use of standard machine learning techniques; and on the other hand, because the MNAR data can introduce large bias in the statistical analysis of databases.

Because of the PPCA hypothesis and the processing of informative missing data, this work has a wide range of applications.

## Footnotes

[1] Note that it implies that $d < p - r$.

[2] It does not require that the linear combination coefficients are non-zero.

[3] For a given set of PPCA parameters, the stochasticity comes from the process of drawing 20 times the latent variables, the additive noise and the missing-data pattern.

[4]Note that this method requires to be trained on a complete dataset.

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
