[Supplementary Material · PPCA_MNAR_NeurIPS2020_supp.pdf]

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

For the sake of readability, we first present the proof of Proposition 1 in the case of the toy example presented in Section 3.1 with $p = 3$ and $r = 2$. The proof in the general setting follows.

## A.1 Proof of Proposition 1 in the case of the toy example presented in Section 3.1

Consider the setting of the toy example presented in Section 3.1 with $p = 3$ and $r = 2$. The PPCA model in (1) reads

$$\begin{cases} Y & = (Y_1 \quad Y_2 \quad Y_3) = (\alpha_1 \quad \alpha_2 \quad \alpha_3) + (W_1 \quad W_2)\,B + \epsilon, \\ Y & \sim \mathcal{N}(\alpha, \Sigma), \ \Sigma = B^T B + \sigma^2 I. \end{cases}$$

$Y_2$ and $Y_3$ are assumed to be observed and $Y_1$ is self-masked MNAR, *i.e.*

$$\mathbb{P}(\Omega_1 = 1 | Y_1, Y_2, Y_3; \phi_1) = \mathbb{P}(\Omega_1 = 1 | Y_1; \phi_1) = F_1(\phi_1^0 + \phi_1^1 y_1), \qquad (14)$$

where $F_1$ is strictly monotone with a positive finite support and where $\phi_1 = (\phi_1^0, \phi_1^1)$.

*Proof.* Assume that $(Y, \Omega)$ and $(Y', \Omega')$ have distributions respectively parameterized by $(\alpha, \Sigma, \phi_1)$ and $(\alpha', \Sigma', \phi_1')$. Assume that $Y$ and $Y'$ have the same observed distribution, *i.e.*

$$\mathcal{L}(Y_1, \Omega_1 = 1; \alpha_1, \Sigma_{11}, \phi_1) = \mathcal{L}(Y_1', \Omega_1' = 1; \alpha_1', \Sigma_{11}', \phi_1') \qquad (15)$$

$$\mathcal{L}(Y_1, Y_j, \Omega_1 = 1; \alpha_1, \alpha_j, \Sigma_{(1j)}, \phi_1) = \mathcal{L}(Y_1', Y_j', \Omega_1' = 1; \alpha_1', \alpha_j', \Sigma_{(1j)}', \phi_1') \qquad j \in \{2, 3\}, \quad (16)$$

where $\Sigma_{(1j)}$ is the covariance matrix $\begin{pmatrix} \Sigma_{11} & \Sigma_{1j} \\ \Sigma_{1j} & \Sigma_{jj} \end{pmatrix}$. In order to show that parameters identifiability holds, we need to show that (15) and (16) imply that $\alpha = \alpha'$, $\Sigma = \Sigma'$ and $\phi_1 = \phi_1'$. Then, under a known noise level $\sigma^2$, we prove that $B$ and $B'$ are equal up to a row permutation.

As $(Y_2, Y_3)$ and $(Y_2', Y_3')$ are fully observed, the parameters of the distributions $\mathcal{L}(Y_2)$, $\mathcal{L}(Y_2')$, $\mathcal{L}(Y_3)$, $\mathcal{L}(Y_3')$, $\mathcal{L}(Y_2, Y_3)$ and $\mathcal{L}(Y_2', Y_3')$ are identifiable. It trivially implies that $\alpha_2 = \alpha_2'$, $\Sigma_{22} = \Sigma_{22}'$, $\alpha_3 = \alpha_3'$, $\Sigma_{33} = \Sigma_{33}'$ and $\Sigma_{23} = \Sigma_{23}'$.

**Identifiability of the MNAR variable variance.**   Equation (15) can be rewritten in terms of density function as follows

$$f_{Y_1, \Omega_1 = 1}(y_1; \alpha_1, \Sigma_{11}, \phi_1) = f_{Y_1', \Omega_1' = 1}(y_1; \alpha_1', \Sigma_{11}', \phi_1') \qquad \forall y_1 \in \mathbb{R}.$$

Given the missing mechanism in (14) and that $Y_{.1} \sim \mathcal{N}(\alpha_1, \Sigma_{11})$, [15, Theorem 1 a)] ensures that $\Sigma_{11} = \Sigma_{11}'$.

**Identifiability of the Mean and the MNAR mechanism parameter.**   Using (15) and (16), the previous computations entail that

$$\mathcal{L}(Y_2 | Y_1, \Omega_1 = 1; \alpha_1, \alpha_2, \Sigma_{(12)}, \phi_1) = \mathcal{L}(Y_2' | Y_1', \Omega_1' = 1; \alpha_1', \alpha_2', \Sigma_{(12)}', \phi_1'),$$

noting that

$$f_{Y_2 | Y_1 = y_1, \Omega_1 = 1}(y_2; \alpha_1, \alpha_2, \Sigma_{(12)}, \phi_1) = \frac{f_{Y_1, Y_2, \Omega_1 = 1}(y_1, y_2; \alpha_1, \alpha_2, \Sigma_{(12)}, \phi_1)}{f_{Y_1, \Omega_1 = 1}(y_1; \alpha_1, \Sigma_{11}, \phi_1)} \qquad \forall (y_1, y_2) \in \mathbb{R}^2$$

One obtains

$$\frac{\mathbb{P}(\Omega_1 = 1 | Y_1 = y_1, Y_2 = y_2; \phi_1) f_{Y_2 | Y_1 = y_1}(y_2; \alpha_1, \alpha_2, \Sigma_{(12)})}{\mathbb{P}(\Omega_1 = 1 | Y_1 = y_1; \phi_1)}$$

$$= \frac{\mathbb{P}(\Omega_1' = 1 | Y_1' = y_1, Y_2' = y_2; \phi_1') f_{Y_2' | Y_1' = y_1}(y_2; \alpha_1', \alpha_2', \Sigma_{(12)}')}{\mathbb{P}(\Omega_1' = 1 | Y_1 = y_1; \phi_1')} \qquad \forall (y_1, y_2) \in \mathbb{R}^2$$

Yet,

$$\begin{aligned} \mathbb{P}(\Omega_1 = 1 | Y_1 = y_1, Y_2 = y_2; \phi_1) &= \mathbb{E}[\mathbb{E}[1_{\Omega_1 = 1} | Y_1 = y_1, Y_2 = y_2, Y_3 = y_3; \phi_1] | Y_1 = y_1, Y_2 = y_2] \\ &= \mathbb{E}[\mathbb{P}(\Omega_1 = 1 | Y = y; \phi_1) | Y_1 = y_1, Y_2 = y_2] \\ &= \mathbb{E}[\mathbb{P}(\Omega_1 = 1 | Y_1 = y_1; \phi_1) | Y_1 = y_1, Y_2 = y_2] \\ &= \mathbb{P}(\Omega_1 = 1 | Y = y_1; \phi_1) \qquad (17) \end{aligned}$$

by measurability. It implies for all $y_1 \in \mathbb{R}$ and $y_2 \in \mathbb{R}$

$$f_{Y_2|Y_1=y_1}(y_2; \alpha_1, \alpha_2, \Sigma_{(12)}) = f_{Y_2'|Y_1'=y_1}(y_2; \alpha_1', \alpha_2', \Sigma_{(12)}')$$

which leads to the equality of the conditional expectations and variances associated to the above densities:

$$\alpha_2 + \Sigma_{12}\Sigma_{11}^{-1}(\alpha_1 - y_1) = \alpha_2 + \Sigma_{12}'\Sigma_{11}^{-1}(\alpha_1' - y_1) \qquad \forall y_1 \in \mathbb{R}$$
$$\Sigma_{22} - \Sigma_{12}^2\Sigma_{11}^{-1} = \Sigma_{22} - (\Sigma_{12}')^2\Sigma_{11}^{-1}.$$

It implies that

$$\Sigma_{12}^2 = (\Sigma_{12}')^2 \implies |\Sigma_{12}| = |\Sigma_{12}'| \tag{18}$$

$$\frac{\Sigma_{21}}{\Sigma_{21}'} = \frac{(\alpha_1' - y_1)}{(\alpha_1 - y_1)} \implies |\alpha_1 - y_1| = |\alpha_1' - y_1| \qquad \forall y_1 \in \mathbb{R} \tag{19}$$

Equation (19) implies that $\alpha_1 = \alpha_1'$, since for $y_1 = \alpha_1'$, one has $\alpha_1 - \alpha_1' = 0$.

Using (16), one has

$$\mathbb{P}(\Omega_1 = 1|Y_1 = y_1, Y_2 = y_2; \phi_1)f_{(Y_1,Y_2)}(y_1, y_2; \alpha_1, \alpha_2, \Sigma_{(12)})$$
$$= \mathbb{P}(\Omega_1' = 1|Y_1' = y_1, Y_2' = y_2; \phi_1')f_{(Y_1',Y_2')}(y_1, y_2; \alpha_1', \alpha_2', \Sigma_{(12)}') \qquad \forall(y_1, y_2) \in \mathbb{R}^2 \tag{20}$$

Using (17),

$$\frac{\exp\left(-\frac{1}{2}\begin{pmatrix} y_1 - \alpha_1 & y_2 - \alpha_2 \end{pmatrix}\Sigma_{(12)}^{-1}\begin{pmatrix} y_1 - \alpha_1 \\ y_2 - \alpha_2 \end{pmatrix}\right)}{\exp\left(-\frac{1}{2}\begin{pmatrix} y_1 - \alpha_1 & y_2 - \alpha_2 \end{pmatrix}(\Sigma_{(12)}')^{-1}\begin{pmatrix} y_1 - \alpha_1 \\ y_2 - \alpha_2 \end{pmatrix}\right)} \frac{\mathbb{P}(\Omega_1 = 1|Y_1 = y_1; \phi_1)}{\mathbb{P}(\Omega_1' = 1|Y_1' = y_1; \phi_1')} = \frac{\sqrt{\det(\Sigma_{(12)})}}{\sqrt{\det(\Sigma_{(12)}')}},$$

where $\det(\Sigma_{(12)})$ denotes the determinant of the matrix $\Sigma_{(12)}$.

With (18), one has $\Sigma_{11}\Sigma_{22} - \Sigma_{12}^2 = \Sigma_{11}\Sigma_{22} - (\Sigma_{12}')^2$ and $\frac{\sqrt{\det(\Sigma_{(12)})}}{\sqrt{\det(\Sigma_{(12)}')}} = 1$.

It leads to $\forall(y_1, y_2) \in \mathbb{R}^2$,

$$K \cdot \frac{\mathbb{P}(\Omega_1 = 1|Y_1 = y_1; \phi_1)}{\mathbb{P}(\Omega_1' = 1|Y_1' = y_1; \phi_1')} = 1,$$

with

$$K := \frac{\exp\left(-\frac{1}{2\det(\Sigma_{(12)})}\left((y_1 - \alpha_1)^2\Sigma_{11} + (y_2 - \alpha_2)^2\Sigma_{22} - 2(y_1 - \alpha_1)(y_2 - \alpha_2)\Sigma_{12}\right)\right)}{\exp\left(-\frac{1}{2\det(\Sigma_{(12)})}\left((y_1 - \alpha_1)^2\Sigma_{11} + (y_2 - \alpha_2)^2\Sigma_{22} - 2(y_1 - \alpha_1')(y_2 - \alpha_2)\Sigma_{12}'\right)\right)}.$$

The quantity $K$ is equal to one, because

$$(y_2 - \alpha_2)\left((y_1 - \alpha_1)\Sigma_{12} - (y_1 - \alpha_1')\Sigma_{12}'\right) = 0$$

using (19). Thus,

$$\frac{\mathbb{P}(\Omega_1 = 1|Y_1 = y_1; \phi_1)}{\mathbb{P}(\Omega_1' = 1|Y_1' = y_1; \phi_1')} = 1 \iff F_1(\phi_1^0 + \phi_1^1 y_1) = F_1((\phi')_1^0 + (\phi')_1^1 y_1) \qquad \forall y_1 \in \mathbb{R}$$

As $F_1$ is strictly monotone, it is an injective function. Thus,

$$\phi_1^0 + \phi_1^1 y_1 = (\phi')_1^0 + (\phi')_1^1 y_1 \qquad \forall y_1 \in \mathbb{R} \iff (\phi_1^0 - (\phi')_1^0) + ((\phi')_1^1 - \phi_1^1)y_1 = 0 \qquad \forall y_1 \in \mathbb{R}$$

It implies $\phi_1 = \phi_1'$.

**Identifiability of the Covariances of the MNAR variable.** Equation (20) thus leads to

$$f_{(Y_1,Y_2)}(y_1, y_2; \alpha_1, \alpha_2, \Sigma_{(12)}) = f_{(Y_1',Y_2')}(y_1, y_2; \alpha_1', \alpha_2', \Sigma_{(12)}') \qquad \forall (y_1, y_2) \in \mathbb{R}^2$$

One can conclude that $\Sigma_{12} = \Sigma_{12}'$. The same reasoning may be done for the covariance between $Y_1$ and $Y_3$.

**Identifiability of the loading matrix.** One wants to prove $B = B'$ up to row permutation. One has

$$\Sigma = \Sigma' \Leftrightarrow \Sigma - \sigma^2 I_{p\times p} = \Sigma' - \sigma^2 I_{p\times p}$$
$$\Leftrightarrow B^T B = (B')^T B' \tag{21}$$

As $B^T B$ is a positive symetric matrix of rank 2, one has the following singular value decomposition,

$$B^T B = (B')^T B' = UDU^T,$$

where $U = (u_1|u_2|u_3) \in \mathbb{R}^{3\times 3}$ the orthogonal matrix of singular vector and

$$D = \begin{pmatrix} \sqrt{d_1} & 0 & 0 \\ 0 & \sqrt{d_2} & 0 \\ 0 & 0 & 0 \end{pmatrix} \in \mathbb{R}^{3\times 3}$$

with $d_1 \geqslant d_2 \geqslant 0$. One can choose

$$B = \begin{pmatrix} \sqrt{d_1} u_1^T \\ \sqrt{d_2} u_2^T \end{pmatrix}$$

noting that a row permutation of B would not change the product $B^T B$. Therefore, $B = B'$ up to a row permutation.

$\square$

## A.2 Proof of Proposition 1 in the general case

We present the proof of Proposition 1 in the general case where $d$ variables are self-masked MNAR and $p - d$ variables are MCAR.

*Proof.* Assume that $(Y, \Omega)$ and $(Y', \Omega')$ have distributions respectively parameterized by $(\alpha, \Sigma, \phi)$ and $(\alpha', \Sigma', \phi')$. Assume that $Y$ and $Y'$ have the same following observed distributions

$$\mathcal{L}(Y_j, \Omega_j = 1; \alpha_j, \Sigma_{jj}, \phi_j) = \mathcal{L}(Y_j', \Omega_j' = 1; \alpha_j', \Sigma_{jj}', \phi_j') \qquad \forall j \in \{1, \ldots, p\}, \tag{22}$$

$$\mathcal{L}(Y_j, Y_k, \Omega_j = 1, \Omega_k = 1; \alpha_j, \alpha_k, \Sigma_{(jk)}, \phi_j, \phi_k)$$
$$= \mathcal{L}(Y_j', Y_k', \Omega_j' = 1, \Omega_k' = 1; \alpha_j', \alpha_k', \Sigma_{(jk)}', \phi_j', \phi_k') \qquad \forall j \neq k \in \{1, \ldots, p\}, \tag{23}$$

where $\Sigma_{(jk)}$ denotes the covariance matrix $\begin{pmatrix} \Sigma_{jj} & \Sigma_{jk} \\ \Sigma_{jk} & \Sigma_{kk} \end{pmatrix}$.

In order to show that parameters identifiability holds, we need to show that (22) and (23) implies that $\alpha = \alpha'$, $\Sigma = \Sigma'$ and $\phi = \phi'$. Then, under a known noise level $\sigma^2$, we will prove that $B$ and $B'$ are equal up to row permutations.

In what follows, $f_{Y_{.j}}$ or $f_{(Y_{.j}, Y_{.k})}$ respectively denote the density function of $Y_{.j}$, and of $(Y_{.j}, Y_{.k})$.

In the following, we will use the following tip, for any $l \in \{1, \ldots, p\}$ and $\mathcal{K} \subset \{1, \ldots, p\}\backslash\{l\}$ such that $0 \leqslant |\mathcal{K}| \leqslant p - 1$,

$$\mathbb{P}(\Omega_l = 1 | Y_l = y_l, Y_{\mathcal{K}} = y_{\mathcal{K}}; \phi_l) = \mathbb{E}[\mathbb{E}[\mathbf{1}_{\Omega_l = 1} | Y; \phi_l] | Y_l = y_l, Y_{\mathcal{K}} = y_{\mathcal{K}}]$$
$$= \mathbb{E}[\mathbb{P}(\Omega_l = 1 | Y = y; \phi_l) | Y_l = y_l, Y_{\mathcal{K}} = y_{\mathcal{K}}]$$

Thus, using the mechanisms in **A01.**,

$$\mathbb{P}(\Omega_l = 1 | Y_l = y_l, Y_{\mathcal{K}} = y_{\mathcal{K}}; \phi_l)$$
$$= \begin{cases} \mathbb{E}[\mathbb{P}(\Omega_l = 1 | Y_l = y_l; \phi_l) | Y_l = y_l, Y_{\mathcal{K}} = y_{\mathcal{K}}] & \text{if } Y_l \text{ is self-masked MNAR} \\ \mathbb{E}[\mathbb{P}(\Omega_l = 1; \phi_l) | Y_l = y_l, Y_{\mathcal{K}} = y_{\mathcal{K}}] & \text{if } Y_l \text{ is MCAR} \end{cases}$$

Thus,

$$\mathbb{P}(\Omega_l = 1 | Y_l = y_l, Y_{\mathcal{K}} = y_{\mathcal{K}}; \phi_l) = \begin{cases} \mathbb{P}(\Omega_l = 1 | Y_l = y_l; \phi_l) & \text{if } Y_l \text{ is self-masked MNAR} \quad (24) \\ \mathbb{P}(\Omega_l = 1; \phi_l) & \text{if } Y_l \text{ is MCAR} \quad (25) \end{cases}$$

by measurability if $Y_l$ is self-masked MNAR and by independence if $Y_l$ is MCAR.

**Identifiability of the parameters for the not-MNAR variables $(Y_j)_{j \in \overline{\mathcal{M}}}$.**

**Mechanism parameter, Mean and Variance of $Y_j, j \in \overline{\mathcal{M}}$.** Equation (22) leads to

$$\mathbb{P}(\Omega_j = 1 | Y_j = y_j; \phi_j) f_{Y_j}(y_j; \alpha_j, \Sigma_{jj}) = \mathbb{P}(\Omega_j' = 1 | Y_j' = y_j; \phi_j') f_{Y_j'}(y_j; \alpha_j', \Sigma_{jj}') \qquad \forall y_j \in \mathbb{R}.$$

Using (25), $P(\Omega_j = 1) = \mathbb{P}(\Omega_j = 1 | Y_j = y_j; \phi_j) = F_j(\phi_j)$. This distribution is identifiable since it pertains to a conditional distribution of the observed data. As $F_j$ is strictly monotone, it implies that

$$F_j(\phi_j) = F_j(\phi_j') \iff \phi_j = \phi_j'.$$

As $\phi_j = \phi_j'$, one obtains

$$f_{Y_j}(y_j; \alpha_j, \Sigma_{jj}) = f_{Y_j'}(y_j; \alpha_j', \Sigma_{jj}') \qquad \forall y_j \in \mathbb{R}$$

which directly implies that $\alpha_j = \alpha_j'$ and $\Sigma_{jj} = \Sigma_{jj}'$, since $Y_j$ and $Y_j'$ are Gaussian variables.

**Covariance between two not MNAR variables $Y_j$ and $Y_k$, $j \neq k \in \overline{\mathcal{M}}$.** Equation (23) gives that for all $(y_j, y_k) \in \mathbb{R}^2$

$$\mathbb{P}(\Omega_j = 1, \Omega_k = 1 | Y_j = y_j, Y_k = y_k; \phi_j, \phi_k) f_{(Y_j, Y_k)}(y_j, y_k; \alpha_j, \alpha_k, \Sigma_{(j,k)})$$
$$= \mathbb{P}(\Omega_j' = 1, \Omega_k' = 1 | Y_j' = y_j, Y_k' = y_k; \phi_j', \phi_k') f_{(Y_j', Y_k')}(y_j, y_k; \alpha_j', \alpha_k', \Sigma_{(j,k)}'), \quad (26)$$

and one has as well that

$$\mathbb{P}(\Omega_j = 1, \Omega_k = 1 | Y_j = y_j, Y_k = y_k; \phi_j, \phi_k) = \mathbb{P}(\Omega_j = 1 | Y_j = y_j; \phi_j) \mathbb{P}(\Omega_k = 1 | Y_k = y_k; \phi_k),$$

using **A02.**. Likewise,

$$\mathbb{P}(\Omega_j' = 1, \Omega_k' = 1 | Y_j' = y_j, Y_k' = y_k; \phi_j', \phi_k') = \mathbb{P}(\Omega_j' = 1 | Y_j' = y_j; \phi_j') \mathbb{P}(\Omega_k' = 1 | Y_k' = y_k; \phi_k').$$

Given that $\phi_j = \phi_j'$ and $\phi_k = \phi_k'$, one obtains

$$\mathbb{P}(\Omega_j = 1, \Omega_k = 1 | Y_j = y_j, Y_k = y_k; \phi_j, \phi_k) = \mathbb{P}(\Omega_j' = 1, \Omega_k' = 1 | Y_j' = y_j, Y_k' = y_k; \phi_j, \phi_k).$$

Thus, Equation (26) leads to, for all $(y_j, y_k) \in \mathbb{R}^2$,

$$f_{(Y_j, Y_k)}(y_j, y_k; \alpha_j, \alpha_k, \Sigma_{(j,k)}) = f_{(Y_j', Y_k')}(y_j, y_k; \alpha_j', \alpha_k', \Sigma_{(j,k)}'),$$

and $\Sigma_{jk} = \Sigma_{jk}'$.

**Identifiability of the parameters for the MNAR variables.**

**Variance of $Y_m, m \in \mathcal{M}$.** Equation (22) gives that

$$f_{(Y_m, \Omega_m = 1)}(y_m; \alpha_m, \Sigma_{mm}, \phi_m) = f_{(Y_m', \Omega_m' = 1)}(y_m; \alpha_m', \Sigma_{mm}', \phi_m') \qquad \forall y_m \in \mathbb{R}.$$

Given the self-masked missing mechanism in **A01.** and that $Y_m \sim \mathcal{N}(\alpha_m, \Sigma_{mm})$, [15, Theorem 1 a)] ensures that $\Sigma_{mm} = \Sigma_{mm}'$.

**Mean and mechanism parameter of $Y_m$, $m \in \mathcal{M}$.** Let $j \in \overline{\mathcal{M}}$ (a not MNAR variable). One has

$$\mathcal{L}(Y_j, \Omega_j = 1 | Y_m, \Omega_m = 1; \alpha_j, \alpha_m, \Sigma_{(jm)}, \phi_j, \phi_m)$$
$$= \mathcal{L}(Y_j', \Omega_j' = 1 | Y_m', \Omega_m' = 1; \alpha_j', \alpha_m', \Sigma_{(jm)}', \phi_j', \phi_m') \quad (27)$$

using (22) and (23) and noting that

$$f_{(Y_j, \Omega_j=1)|Y_m=y_m, \Omega_m=1}(y_j; \alpha_j, \alpha_m, \Sigma_{(jm)}, \phi_j, \phi_m)$$
$$= \frac{f_{(Y_j, \Omega_j=1, Y_m, \Omega_m=1)}(y_j, y_m; \alpha_j, \alpha_m, \Sigma_{(jm)}, \phi_j, \phi_m)}{f_{(Y_m, \Omega_m=1)}(y_m; \alpha_m, \Sigma_{mm}, \phi_m)} \qquad \forall (y_j, y_m) \in \mathbb{R}^2.$$

Equation (27) implies that $\forall (y_j, y_m) \in \mathbb{R}^2$,

$$\mathbb{P}(\Omega_j = 1 | Y_j = y_j, Y_m = y_m, \Omega_m = 1; \phi_j) \frac{\mathbb{P}(\Omega_m = 1 | Y_j = y_j, Y_m = y_m; \phi_m) f_{Y_j|Y_m=y_m}(y_j; \alpha_j, \alpha_m, \Sigma_{(jm)})}{\mathbb{P}(\Omega_m = 1 | Y_m = y_m; \phi_m)}$$
$$= \mathbb{P}(\Omega_j' = 1 | Y_j' = y_j, Y_m' = y_m, \Omega_m' = 1; \phi_j') \frac{\mathbb{P}(\Omega_m' = 1 | Y_j' = y_j, Y_m' = y_m; \phi_m') f_{Y_j'|Y_m'=y_m}(y_j; \alpha_j', \alpha_m', \Sigma_{(jm)}')}{\mathbb{P}(\Omega_m' = 1 | Y_m' = y_m; \phi_m')}$$
$$(28)$$

One can note that

$$\mathbb{P}(\Omega_j = 1 | Y_j = y_j, Y_m = y_m, \Omega_m = 1; \phi_j) = \mathbb{P}(\Omega_j = 1 | Y_j = y_j; \phi_j).$$

Indeed,

$$\mathbb{P}(\Omega_j = 1 | Y_j = y_j, Y_m = y_m, \Omega_m = 1; \phi_j) = \frac{\mathbb{P}(\Omega_j = 1 \cap \Omega_m = 1 | Y_j = y_j, Y_m = y_m; \phi_j, \phi_m)}{\mathbb{P}(\Omega_m = 1 | Y_j = y_j, Y_m = y_m; \phi_m)}$$
$$= \frac{\mathbb{P}(\Omega_j = 1 | Y_j = y_j; \phi_j) \mathbb{P}(\Omega_m = 1 | Y_m = y_m; \phi_m)}{\mathbb{P}(\Omega_m = 1 | Y_j = y_j, Y_m = y_m; \phi_m)}$$
$$= \mathbb{P}(\Omega_j = 1 | Y_j = y_j; \phi_j),$$

using **A02.** in the second step. Likewise,

$$\mathbb{P}(\Omega_j' = 1 | Y_j' = y_j, Y_m' = y_m, \Omega_m' = 1; \phi_j') = \mathbb{P}(\Omega_j' = 1 | Y_j' = y_j; \phi_j').$$

Given that $\phi_j = \phi_j'$,

$$\mathbb{P}(\Omega_j = 1 | Y_j = y_j, Y_m = y_m, \Omega_m = 1; \phi_j) = \mathbb{P}(\Omega_j' = 1 | Y_j' = y_j, Y_m' = y_m, \Omega_m' = 1; \phi_j')$$

Thus, Equation (28) leads to

$$\frac{\mathbb{P}(\Omega_m = 1 | Y_j = y_j, Y_m = y_m; \phi_m) f_{Y_j|Y_m=y_m}(y_j; \alpha_j, \alpha_m, \Sigma_{(jm)})}{\mathbb{P}(\Omega_m = 1 | Y_m = y_m; \phi_m)}$$
$$= \frac{\mathbb{P}(\Omega_m' = 1 | Y_j' = y_j, Y_m' = y_m; \phi_m') f_{Y_j'|Y_m'=y_m}(y_j; \alpha_j', \alpha_m', \Sigma_{(jm)}')}{\mathbb{P}(\Omega_m' = 1 | Y_m' = y_m; \phi_m')} \qquad \forall (y_j, y_m) \in \mathbb{R}^2.$$

As $\mathbb{P}(\Omega_m = 1 | Y_j = y_j, Y_m = y_m; \phi_m) = \mathbb{P}(\Omega_m = 1 | Y_m = y_m; \phi_m)$ by using (A.2), one obtains

$$f_{Y_j|Y_m=y_m}(y_j; \alpha_j, \alpha_m, \Sigma_{(jm)}) = f_{Y_j'|Y_m'=y_m}(y_j; \alpha_j', \alpha_m', \Sigma_{(jm)}') \qquad \forall (y_j, y_m) \in \mathbb{R}^2,$$

which leads to the equality of the conditional expectation and variance, as follows:

$$\alpha_j + \Sigma_{mj} \Sigma_{mm}^{-1} (\alpha_m - y_m) = \alpha_j' + \Sigma_{mj}' (\Sigma_{mm}')^{-1} (\alpha_m' - y_m) \qquad \forall (y_j, y_m) \in \mathbb{R}^2$$
$$\Sigma_{jj} - \Sigma_{mj}^2 \Sigma_{mm}^{-1} = \Sigma_{jj}' - (\Sigma_{mj}')^2 (\Sigma_{mm}')^{-1}$$

As $\alpha_j = \alpha_j'$ and $\Sigma_{mm} = \Sigma_{mm}'$,

$$\Sigma_{mj}^2 = (\Sigma_{mj}')^2 \implies |\Sigma_{mj}| = |\Sigma_{mj}'| \tag{29}$$
$$\frac{\Sigma_{mj}}{\Sigma_{mj}'} = \frac{(\alpha_m' - y_m)}{(\alpha_m - y_m)} \implies |\alpha_m - y_m| = |\alpha_m' - y_m| \qquad \forall y_m \in \mathbb{R} \tag{30}$$

Equation (30) implies that $\alpha_m = \alpha'_m$, since for $y_m = \alpha'_m$, one has $\alpha_m - \alpha'_m = 0$.

In addition, using (23), one has for all $(y_j, y_m) \in \mathbb{R}^2$,

$$\mathbb{P}(\Omega_j = 1, \Omega_m = 1 | Y_j = y_j, Y_m = y_m; \phi_j, \phi_m) f_{(Y_j, Y_m)}(y_j, y_m; \alpha_j, \alpha_m, \Sigma_{(jm)})$$
$$= \mathbb{P}(\Omega'_j = 1, \Omega'_m = 1 | Y'_j = y_j, Y'_m = y_m; \phi'_j, \phi'_m) f_{(Y'_j, Y'_m)}(y_j, y_m; \alpha'_j, \alpha'_m, \Sigma'_{(jm)}) \quad (31)$$

One can note that

$$\mathbb{P}(\Omega_j = 1, \Omega_m = 1 | Y_j = y_j, Y_m = y_m; \phi_j, \phi_m)$$
$$= \mathbb{P}(\Omega_j = 1; \phi_j) \mathbb{P}(\Omega_m = 1 | Y_m = y_m; \phi_m),$$

using **A02.** and the tips given in and (25). The same equation holds for $(Y'_j, Y'_m, \Omega'_j, \Omega'_m)$ with the parameters $(\phi'_j, \phi'_m)$. Using $\phi_j = \phi'_j$, Equation (31) leads to

$$\mathbb{P}(\Omega_m = 1 | Y_m = y_m; \phi_m) f_{(Y_j, Y_m)}(y_j, y_m; \alpha_j, \alpha_m, \Sigma_{(jm)}) =$$
$$\mathbb{P}(\Omega'_m = 1 | Y'_m = y_m; \phi'_m) f_{(Y'_j, Y'_m)}(y_j, y_m; \alpha'_j, \alpha'_m, \Sigma'_{(jm)}) \quad \forall (y_j, y_m) \in \mathbb{R}^2. \quad (32)$$

It implies that, $\forall (y_j, y_m) \in \mathbb{R}^2$,

$$\frac{\exp\left(-\frac{1}{2} \begin{pmatrix} y_j - \alpha_j & y_m - \alpha_m \end{pmatrix} \Sigma_{(jm)}^{-1} \begin{pmatrix} y_j - \alpha_j \\ y_m - \alpha_m \end{pmatrix}\right)}{\exp\left(-\frac{1}{2} \begin{pmatrix} y_j - \alpha'_j & y_m - \alpha'_m \end{pmatrix} (\Sigma'_{(jm)})^{-1} \begin{pmatrix} y_j - \alpha'_j \\ y_m - \alpha'_m \end{pmatrix}\right)} \frac{\mathbb{P}(\Omega_m = 1 | Y_m = y_m; \phi_m)}{\mathbb{P}(\Omega'_m = 1 | Y'_m = y_m; \phi'_m)} = \frac{\sqrt{\det(\Sigma_{(jm)})}}{\sqrt{\det(\Sigma'_{(jm)})}},$$

where $\det(\Sigma_{(jm)})$ denotes the determinant of the covariance matrix $\Sigma_{(jm)}$.

With $\Sigma_{jj} = \Sigma'_{jj}$, $\Sigma_{mm} = \Sigma'_{mm}$ and Equation (29), one has

$$\Sigma_{jj} \Sigma_{mm} - \Sigma_{mj}^2 = \Sigma_{jj} \Sigma_{mm} - (\Sigma'_{mj})^2 \quad \implies \quad \frac{\sqrt{\det(\Sigma_{(jm)})}}{\sqrt{\det(\Sigma'_{(jm)})}} = 1.$$

Besides, using $\alpha_j = \alpha'_j$, $\Sigma_{jj} = \Sigma'_{jj}$ and $\Sigma_{mm} = \Sigma'_{mm}$, one obtains that for all $(y_j, y_m) \in \mathbb{R}^2$,

$$K \cdot \frac{\mathbb{P}(\Omega_m = 1 | Y_m = y_m; \phi_m)}{\mathbb{P}(\Omega'_m = 1 | Y'_m = y_m; \phi'_m)} = 1,$$

with

$$K := \frac{\exp\left(-\frac{1}{2\det(\Sigma_{(jm)})} \left((y_j - \alpha_j)^2 \Sigma_{jj} + (y_m - \alpha_m)^2 \Sigma_{mm} - 2(y_j - \alpha_j)(y_m - \alpha_m)\Sigma_{mj}\right)\right)}{\exp\left(-\frac{1}{2\det(\Sigma_{(jm)})} \left((y_j - \alpha_j)^2 \Sigma_{jj} + (y_m - \alpha_m)^2 \Sigma_{mm} - 2(y_j - \alpha_j)(y_m - \alpha'_m)\Sigma'_{mj}\right)\right)}.$$

The quantity $K$ is equal to one, because

$$(y_j - \alpha_j)((y_m - \alpha_m)\Sigma_{mj} - (y_m - \alpha'_m)\Sigma'_{mj}) = 0$$

using (30). Thus, for all $y_m \in \mathbb{R}$,

$$\frac{\mathbb{P}(\Omega_m = 1 | Y_m = y_m; \phi_m)}{\mathbb{P}(\Omega'_m = 1 | Y'_m = y_m; \phi'_m)} = 1 \quad \Longleftrightarrow \quad F_m(\phi_m^0 + \phi_m^1 y_m) = F_m((\phi')_m^0 + (\phi')_m^1 y_m).$$

As F is strictly monotone, it is an injective function. Thus,

$$\phi_m^0 + \phi_m^1 y_m = (\phi')_m^0 + (\phi')_m^1 y_m \Leftrightarrow ((\phi')_m^0 - \phi_m^0) + ((\phi')_m^1 - \phi_m^1)y_m = 0 \quad \forall y_1 \in \mathbb{R}$$

It implies that $\phi_m = \phi'_m$.

**Covariance between $Y_j$ and $Y_m$ with $j \in \overline{\mathcal{M}}, m \in \mathcal{M}$.** Using (32) and $\phi_m = \phi'_m$, one has

$$f_{(Y_j, Y_m)}(y_j, y_m; \alpha_j, \alpha_m, \Sigma_{(jm)}) = f_{(Y'_j, Y'_m)}(y_j, y_m; \alpha'_j, \alpha'_m, \Sigma'_{(jm)}) \quad \forall (y_j, y_m) \in \mathbb{R}^2$$

One can conclude that $\Sigma_{mj} = \Sigma'_{mj}$.

**Covariance between $Y_\ell$ and $Y_m$ with $\ell \neq m \in \mathcal{M}$.** Using (23), one has for all $(y_\ell, y_m) \in \mathbb{R}^2$,

$$\mathbb{P}(\Omega_\ell = 1, \Omega_m = 1|Y_j = y_j, Y_m = y_m; \phi_\ell, \phi_m)f_{(Y_\ell, Y_m)}(y_\ell, y_m; \alpha_\ell, \alpha_m, \Sigma_{(\ell m)})$$
$$= \mathbb{P}(\Omega'_\ell = 1, \Omega'_m = 1|Y'_\ell = y_\ell, Y'_m = y_m; \phi'_\ell, \phi'_m)f_{(Y'_\ell, Y'_m)}(y_\ell, y_m; \alpha'_\ell, \alpha'_m, \Sigma'_{(\ell m)}) \quad (33)$$

One can note that

$$\mathbb{P}(\Omega_\ell = 1, \Omega_m = 1|Y_\ell = y_\ell, Y_m = y_m; \phi_\ell, \phi_m)$$
$$= \mathbb{P}(\Omega_\ell = 1|Y_\ell = y_\ell; \phi_\ell)\mathbb{P}(\Omega_m = 1|Y_m = y_m; \phi_m),$$

using **A02.** and the tip given in (A.2). The same equation holds for $(Y'_\ell, Y'_m, \Omega'_\ell, \Omega'_m)$ with the parameters $(\phi'_\ell, \phi'_m)$. Yet $\phi_\ell = \phi'_\ell$ and $\phi_m = \phi'_m$, which gives, for all $(y_j, y_m) \in \mathbb{R}^2$,

$$\mathbb{P}(\Omega_\ell = 1, \Omega_m = 1|Y_\ell = y_\ell, Y_m = y_m; \phi_\ell, \phi_m) = \mathbb{P}(\Omega'_\ell = 1, \Omega'_m = 1|Y'_\ell = y_\ell, Y'_m = y_m; \phi_\ell, \phi'_m).$$

Equation (33) leads to

$$f_{(Y_\ell, Y_m)}(y_\ell, y_m; \alpha_\ell, \alpha_m, \Sigma_{(\ell m)}) = f_{(Y'_\ell, Y'_m)}(y_\ell, y_m; \alpha'_\ell, \alpha'_m, \Sigma'_{(\ell m)}) \qquad \forall (y_\ell, y_m) \in \mathbb{R}^2,$$

which implies that $\Sigma_{\ell m} = \Sigma'_{\ell m}$.

**Identifiability of the loading matrix.** One wants to prove that $B = B'$ up to a row permutation. One has

$$\Sigma = \Sigma' \iff \Sigma - \sigma^2 I_{p \times p} = \Sigma' - \sigma^2 I_{p \times p}$$
$$\iff B^T B = (B')^T B' \quad (34)$$

As $B^T B$ is a positive symetric matrix of rank $r$, its singular value decomposition reads

$$B^T B = (B')^T B' = UDU^T,$$

where $U = (u_1| \dots |u_p) \in \mathbb{R}^{p \times p}$ is an orthogonal matrix containing the singular vectors and

$$D = \begin{pmatrix} \sqrt{d_1} & & & & & \\ & \ddots & & & 0 & \\ & & \sqrt{d_r} & & & \\ & 0 & & 0 & & \\ & & & & \ddots & \\ & & & & & 0 \end{pmatrix} \in \mathbb{R}^{p \times p}$$

with $d_1 \geqslant \dots \geqslant d_r \geqslant 0$. One can choose

$$B = \begin{pmatrix} \sqrt{d_1}u_1^T \\ \hline \vdots \\ \hline \sqrt{d_r}u_r^T \end{pmatrix}$$

A row permutation of B does not change the product $B^T B$. Therefore, $B = B'$ up to a row permutation.

$\square$

# B  Proof for Section 3

## B.1  Proof of Lemma 2

**Lemma 2.** *Under the PPCA model* (1) *and Assumption **A1.**, choose $j \in \mathcal{J}$. Denote $B^{-1} \in \mathbb{R}^{r \times r}$ the inverse of $\left( B_{.m} \quad (B_{.j'})_{j' \in \mathcal{J}_{-j}} \right)$. One has*

$$Y_{.j} = \mathcal{B}_{j \to m, \mathcal{J}_{-j}}[0] + \sum_{j' \in \mathcal{J}_{-j}} \mathcal{B}_{j \to m, \mathcal{J}_{-j}}[j']Y_{.j'} + \mathcal{B}_{j \to m, \mathcal{J}_{-j}}[m]Y_{.m} + \zeta$$

*with:*

$$\mathcal{B}_{j \to m, \mathcal{J}_{-j}}[j'] := \sum_{k \in \{m\} \cup \mathcal{J}_{-j}} B_{kj'}^{-1} B_{jk}, \forall j' \in \mathcal{J}_{-j}$$

$$\mathcal{B}_{j \to m, \mathcal{J}_{-j}}[m] := \sum_{k \in \{m\} \cup \mathcal{J}_{-j}} B_{km}^{-1} B_{jk},$$

$$\mathcal{B}_{j \to m, \mathcal{J}_{-j}}[0] := \mathbf{1}\alpha_j - \sum_{j' \in \mathcal{J}_{-j}} \mathcal{B}_{j \to m, \mathcal{J}_{-j}}[j'] \mathbf{1}\alpha_{j'} - \mathcal{B}_{j \to m, \mathcal{J}_{-j}}[m] \mathbf{1}\alpha_m$$

$$\zeta = - \sum_{j' \in \mathcal{J}_{-j}} \mathcal{B}_{j \to m, \mathcal{J}_{-j}}[j'] \epsilon_{.j'} - \mathcal{B}_{j \to m, \mathcal{J}_{-j}}[m] \epsilon_{.m} + \epsilon_{.j}$$

*Proof.* Starting from the PPCA model written in (1) and recalled here

$$Y = \mathbf{1}\alpha + WB + \epsilon$$

and the matrix $B \in \mathbb{R}^{r \times p}$ being of full rank $r$, solving this linear system is the same as solving the following reduced system

$$\begin{pmatrix} Y_{.m} & (Y_{.j'})_{j' \in \mathcal{J}_{-j}} \end{pmatrix} = \mathbf{1}\alpha_{|r} + \begin{pmatrix} W_{.1} & \dots & W_{.r} \end{pmatrix} B_{|r} + \epsilon_{|r},$$

where $B_{|r} \in \mathbb{R}^{r \times r}$ denotes the reduced matrix $\begin{pmatrix} B_{.m} & (B_{.j'})_{j' \in \mathcal{J}_{-j}} \end{pmatrix}$ of $B$. Similarly, $\alpha_{|r} \in \mathbb{R}^r$ and $\epsilon_{|r} \in \mathbb{R}^{n \times r}$ denote the reduced matrices of $\alpha$ and $\epsilon$. With a slight abuse of notation, $B^{-1}$ denotes the inverse of the reduced matrix $\begin{pmatrix} B_{.m} & (B_{.j'})_{j' \in \mathcal{J}_{-j}} \end{pmatrix}$ which exists using **A1.**.

Then, one can derive that

$$\begin{pmatrix} W_{.1} & \dots & W_{.r} \end{pmatrix} = \left( \begin{pmatrix} Y_{.m} & (Y_{.j'})_{j' \in \mathcal{J}_{-j}} \end{pmatrix} - \mathbf{1}\alpha_{|r} - \epsilon_{|r} \right) B^{-1}.$$

The expression of $Y_{.j}$ as a function of the latent variables is

$$Y_{.j} = \mathbf{1}\alpha_j + \begin{pmatrix} W_{.1} & \dots & W_{.r} \end{pmatrix} B_{j.} + \epsilon_{.j}$$
$$= \mathbf{1}\alpha_j + \left( \begin{pmatrix} Y_{.m} & (Y_{.j'})_{j' \in \mathcal{J}_{-j}} \end{pmatrix} - \mathbf{1}\alpha_{|r} - \epsilon_{|r} \right) B^{-1} B_{j.} + \epsilon_{.j},$$

so that

$$Y_{.j} = \sum_{\ell \in \{m\} \cup \mathcal{J}_{-j}} \left( \sum_{k \in \{m\} \cup \mathcal{J}_{-j}} B_{k\ell}^{-1} B_{jk} \right) Y_{.\ell}$$

$$- \sum_{\ell \in \{m\} \cup \mathcal{J}_{-j}} \left( \sum_{k \in \{m\} \cup \mathcal{J}_{-j}} B_{k\ell}^{-1} B_{jk} \right) (\mathbf{1}\alpha_\ell + \epsilon_{.\ell}) + \epsilon_{.j} + \mathbf{1}\alpha_j.$$

which leads to the desired solution.

$\square$

## B.2 Proof of Proposition 4

**Proposition 4** (Mean estimator). *Consider the PPCA model* (1). *Under Assumptions **A1.** and **A2.**, an estimator of the mean of a MNAR variable $Y_{.m}$, for $m \in \mathcal{M}$, can be constructed as follows: choose $j \in \mathcal{J}$, and compute*

$$\hat{\alpha}_m := \frac{\hat{\alpha}_j - \hat{\mathcal{B}}_{j \to m, \mathcal{J}_{-j}}^c[0] - \sum_{j' \in \mathcal{J}_{-j}} \hat{\mathcal{B}}_{j \to m, \mathcal{J}_{-j}}^c[j'] \hat{\alpha}_{j'}}{\hat{\mathcal{B}}_{j \to m, \mathcal{J}_{-j}}^c[m]},$$

*with the $(\hat{\mathcal{B}}_{j \to m, \mathcal{J}_{-j}}[k])$'s estimators of the coefficients given in Definition 3 and assuming that the coefficient $\mathcal{B}_{j \to m, \mathcal{J}_{-j}}^c[m]$ estimated by $\hat{\mathcal{B}}_{j \to m, \mathcal{J}_{-j}}^c[m]$ is non zero.*

*Under the additional Assumptions **A3.** and **A4.**, this estimator is consistent.*

*Proof.* The main goal is to obtain a formula for $\alpha_{.m}$, *i.e.*

$$\alpha_m = \frac{\alpha_j - \mathcal{B}^c_{j \to m, \mathcal{J}_{-j}}[0] - \sum_{j' \in \mathcal{J}_{-j}} \mathcal{B}^c_{j \to m, \mathcal{J}_{-j}}[j'] \alpha_{j'}}{\mathcal{B}^c_{j \to m, \mathcal{J}_{-j}}[m]}, \tag{35}$$

from which an estimator can be deduced. The idea is to express $\alpha_j$ from $\alpha_m$ and $(\alpha_{j'})_{j' \in \mathcal{J}_{-j}}$. Note that $\mathbb{E}[Y_{.j}] = \mathbb{E}[\mathbb{E}[Y_{.j}|(Y_{.k})_{k \in \overline{\{j\}}}]]$. Assumption **A2.** leads to

$$\mathbb{E}[Y_{.j}|(Y_{.k})_{k \in \overline{\{j\}}}] = \mathbb{E}[Y_{.j}|(Y_{.k})_{k \in \overline{\{j\}}}, \Omega_{.m} = 1].$$

Then, by Definition 3 which gives $(Y_{.j})_{|\Omega_{.m}=1}$,

$$\mathbb{E}[Y_{.j}|(Y_{.k})_{k \in \overline{\{j\}}}, \Omega_{.m} = 1]$$

$$= \mathbb{E}\left[ \mathcal{B}^c_{j \to m, \mathcal{J}_{-j}}[0] + \sum_{k \in \{m\} \cup \mathcal{J}_{-j}} \mathcal{B}^c_{j \to m, \mathcal{J}_{-j}}[k] Y_{.k} + \zeta^c \Bigg| (Y_{.k})_{k \in \overline{\{j\}}} \right]$$

$$= \mathcal{B}^c_{j \to m, \mathcal{J}_{-j}}[0] + \sum_{k \in \{m\} \cup \mathcal{J}_{-j}} \mathcal{B}^c_{j \to m, \mathcal{J}_{-j}}[k] Y_{.k} + \mathbb{E}\left[ \zeta^c \Bigg| (Y_{.k})_{k \in \overline{\{j\}}} \right]$$

Thus, by taking the mean and given that $\mathbb{E}[\epsilon_{.k}] = 0, \forall k \in \{m\} \cup \mathcal{J}_{-j}$, one has

$$\alpha_j = \mathcal{B}^c_{j \to m, \mathcal{J}_{-j}}[0] + \sum_{j' \in \mathcal{J}_{-j}} \mathcal{B}^c_{j \to m, \mathcal{J}_{-j}}[j'] \alpha_{j'} + \mathcal{B}^c_{j \to m, \mathcal{J}_{-j}}[m] \alpha_m,$$

implying Equation (35), provided that $\mathcal{B}^c_{j \to m, \mathcal{J}_{-j}}[m] \neq 0$.

From this formula for the mean $\alpha_m$, one define its estimator $\hat{\alpha}_m$ as in (9). It is trivially consistent as the linear combination of consistent quantities under **A3.** and **A4.** $\qquad\square$

## B.3   Proof of Proposition 5

**Proposition 5** (Variance and covariances estimators). *Consider the PPCA model* (1). *Under Assumptions A1. and A2., an estimator of the variance of a MNAR variable $Y_{.m}$ for $m \in \mathcal{M}$ and its covariances with the pivot variables, can be constructed as follows: choose $j \in \mathcal{J}$ and compute*

$$\left( \widehat{\mathrm{Var}}(Y_{.m}) \quad \widehat{\mathrm{Cov}}(Y_{.m}, (Y_{.k})_{k \in \mathcal{J}}) \right)^T := (\widehat{M}_j)^{-1} \widehat{P}_j,$$

*assuming that $\sigma^2$ tends to zero and the inverse of the matrix $M_j$ estimated by $(\widehat{M}_j)^{-1}$ exists, with*

$$\widehat{M}_j = \begin{matrix} \in \mathbb{R} \{ \\ \in \mathbb{R}^r \{ \end{matrix} \begin{bmatrix} \overbrace{(\hat{\mathcal{B}}^c_{j \to m, \mathcal{J}_{-j}}[m])^2}^{\in \mathbb{R}} & \overbrace{0 \quad 2\hat{\mathcal{B}}^c_{j \to m, \mathcal{J}_{-j}}[m] \left( \hat{\mathcal{B}}^c_{j \to m, \mathcal{J}_{-j}}[\mathcal{J}_{-j}] \right)^T}^{\in \mathbb{R}^r} \\ -(\hat{\mathcal{B}}^c_{k \to m, \mathcal{J}_{-k}}[m])_{k \in \mathcal{J}} & (\widehat{M}^k)_{k \in \mathcal{J}} \end{bmatrix}$$

*Let us precise that $\widehat{M}_j \in \mathbb{R}^{(r+1) \times (r+1)}$. One has $(\hat{\mathcal{B}}^c_{k \to m, \mathcal{J}_{-k}}[m])_{k \in \mathcal{J}} = \begin{pmatrix} \hat{\mathcal{B}}^c_{j_1 \to m, \mathcal{J}_{-j_1}}[m] \\ \vdots \\ \hat{\mathcal{B}}^c_{j_r \to m, \mathcal{J}_{-j_r}}[m] \end{pmatrix}$.*

*One details $\widehat{M}^k$ for $k = j_1$ and the same definition is valid for all $k \in \mathcal{J}$.*

$$\widehat{M}^{j_1} = \left( 1 \quad -\hat{\mathcal{B}}^c_{j_1 \to m, \mathcal{J}_{-j_1}}[j_2] \quad \cdots \quad -\hat{\mathcal{B}}^c_{j_1 \to m, \mathcal{J}_{-j_1}}[j_r] \right) \in \mathbb{R}^r$$

$$\widehat{P}_j = \begin{bmatrix} \overbrace{(\widehat{\mathrm{Var}}(Y_{.j}) - Q^c - (\hat{\mathcal{B}}^c_{j \to m, \mathcal{J}_{-j}}[\mathcal{J}_{-j}])^T \widehat{\mathrm{Var}}(Y_{\mathcal{J}_{-j}}) \hat{\mathcal{B}}^c_{j \to m, \mathcal{J}_{-j}}[\mathcal{J}_{-j}]}^{\in \mathbb{R}} \\ \left( ((\hat{\mathcal{B}}^c_{k \to m, \mathcal{J}_{-k}})^T \left( 1 \quad \hat{\alpha}_m \quad (\hat{\alpha}_\ell)_{\ell \in \mathcal{J}_{-k}} \right)^T - \hat{\alpha}_k) \hat{\alpha}_m \right)_{k \in \mathcal{J}} \end{bmatrix} \begin{matrix} \} \in \mathbb{R} \\ \\ \} \in \mathbb{R}^r \end{matrix}$$

$$\hat{Q}^c = \left(\widehat{\mathrm{Var}}(Y_{.j})\big|\Omega_{.m} = 1\right)$$
$$- \left(\widehat{\mathrm{Cov}}((Y_{.k})_{k\in\overline{\{j\}}}, Y_{.j})\widehat{\mathrm{Var}}((Y_{.k})_{k\in\overline{\{j\}}})^{-1}\widehat{\mathrm{Cov}}((Y_{.k})_{k\in\overline{\{j\}}}, Y_{.j})^T\big|\Omega_{.m} = 1\right).$$

*Under the additional Assumptions **A3.** and **A4.**, the estimators for the variance of $Y_{.m}$ and its covariances with the pivot variables given in (11) are consistent.*

*Proof.* As for the mean, to derive some estimator of the variance and the covariances, we want to obtain a formula as

$$M_j \left(\mathrm{Var}(Y_{.m}) \quad \mathrm{Cov}(Y_{.m}, (Y_{.k})_{k\in\mathcal{J}})\right)^T = \left(P_j - \mathcal{O}(\sigma^2)\right), \tag{36}$$

with

$$M_j = \begin{array}{l} \in \mathbb{R} \left\{ \\ \in \mathbb{R}^r \left\{ \right. \end{array} \left[ \begin{array}{ccc} \overbrace{(\mathcal{B}^c_{j\to m, \mathcal{J}_{-j}}[m])^2}^{\in\mathbb{R}} & \overbrace{0 \quad 2\mathcal{B}^c_{j\to m, \mathcal{J}_{-j}}[m]\left(\mathcal{B}^c_{j\to m, \mathcal{J}_{-j}}[\mathcal{J}_{-j}]\right)^T}^{\in\mathbb{R}^r} \\ -(\mathcal{B}^c_{k\to m, \mathcal{J}_{-k}}[m])_{k\in\mathcal{J}} & (M^k)_{k\in\mathcal{J}} \end{array} \right]$$

Let us precise that $M_j \in \mathbb{R}^{(r+1)\times(r+1)}$. One has $(\mathcal{B}^c_{k\to m, \mathcal{J}_{-k}}[m])_{k\in\mathcal{J}} = \begin{pmatrix} \mathcal{B}^c_{j_1\to m, \mathcal{J}_{-j_1}}[m] \\ \vdots \\ \mathcal{B}^c_{j_r\to m, \mathcal{J}_{-j_r}}[m] \end{pmatrix}.$

One details $M^k$ for $k = j_1$ and the same definition is valid for all $k \in \mathcal{J}$.

$$M^{j_1} = \left(1 \quad -\mathcal{B}^c_{j_1\to m, \mathcal{J}_{-j_1}}[j_2] \quad \cdots \quad -\mathcal{B}^c_{j_1\to m, \mathcal{J}_{-j_1}}[j_r]\right) \in \mathbb{R}^r$$

$$P_j = \left[ \begin{array}{c} \overbrace{(\mathrm{Var}(Y_{.j}) - Q^c - (\mathcal{B}^c_{j\to m, \mathcal{J}_{-j}}[\mathcal{J}_{-j}])^T \mathrm{Var}(Y_{\mathcal{J}_{-j}})\mathcal{B}^c_{j\to m, \mathcal{J}_{-j}}[\mathcal{J}_{-j}])}^{\in\mathbb{R}} \\ \left(((\mathcal{B}^c_{k\to m, \mathcal{J}_{-k}})^T \left(1 \quad \mathbb{E}[Y_{.m}] \quad (\mathbb{E}[Y_{.\ell}])_{\ell\in\mathcal{J}_{-k}}\right)^T - \mathbb{E}[Y_{.k}])\mathbb{E}[Y_{.m}]\right)_{k\in\mathcal{J}} \end{array} \right] \begin{array}{l} \Big\}\in\mathbb{R} \\ \Big\}\in\mathbb{R}^r \end{array}$$

$$\mathcal{O}(\sigma^2) = \left[ \begin{array}{c} \overbrace{o_{\mathrm{var}}(\sigma^2)}^{\in\mathbb{R}} \\ -\left(o_{\mathrm{cov},k}(\sigma^2)\right)_{k\in\mathcal{J}} \end{array} \right] \begin{array}{l} \Big\}\in\mathbb{R} \\ \Big\}\in\mathbb{R}^r \end{array},$$

with $o_{\mathrm{var}}(\sigma^2)$ and $o_{\mathrm{cov},k}(\sigma^2)$ detailed in (42) and (45) respectively.

$$Q^c = \left(\mathrm{Var}(Y_{.j})\big|\Omega_{.m} = 1\right)$$
$$- \left(\mathrm{Cov}((Y_{.k})_{k\in\overline{\{j\}}}, Y_{.j})\mathrm{Var}((Y_{.k})_{k\in\overline{\{j\}}})^{-1}\mathrm{Cov}((Y_{.k})_{k\in\overline{\{j\}}}, Y_{.j})^T\big|\Omega_{.m} = 1\right). \tag{37}$$

The strategy is to prove each equality of the linear system in (36).

**Deriving an equation for the variance.** The idea is first to express $\mathrm{Var}(Y_{.j})$ from $\mathrm{Var}(Y_{.m})$, $(\mathrm{Var}(Y_{.j'}))_{j'\in\mathcal{J}_{-j}}$ and $(\mathrm{Cov}(Y_{.k}, Y_{.\ell}))_{k\neq\ell\in\{m\}\cup\mathcal{J}_{-j}}$. The law of total variance reads as

$$\mathrm{Var}(Y_{.j}) = \mathbb{E}[\mathrm{Var}(Y_{.j}|Z)] + \mathrm{Var}(\mathbb{E}[Y_{.j}|Z]), \tag{38}$$

with $Z = (Y_{.k})_{k\in\overline{\{j\}}}$.

For the first term in (38), using Assumption **A2.**, one has

$$Y_{.j} \perp\!\!\!\perp (\Omega_{.m} = 1)|Z$$

which leads to

$$\mathrm{Var}(Y_{.j}|Z) = \mathrm{Var}(Y_{.j}|Z, \Omega_{.m} = 1).$$

The conditional variance for a Gaussian vector gives

$$\mathrm{Var}(Y_{.j}|Z) = \mathrm{Var}(Y_{.j}) - \mathrm{Cov}(Z, Y_{.j})\mathrm{Var}(Z)^{-1}\mathrm{Cov}(Z, Y_{.j})^T,$$

implying that

$$\mathrm{Var}(Y_{.j}|Z, \Omega_{.m} = 1) = \big(\mathrm{Var}(Y_{.j}) - \mathrm{Cov}(Z, Y_{.j})\mathrm{Var}(Z)^{-1}\mathrm{Cov}(Z, Y_{.j})^T\big|\Omega_{.m} = 1\big)$$

and then, as deterministic quantity,

$$\mathbb{E}[\mathrm{Var}(Y_{.j}|Z)] = \big(\mathrm{Var}(Y_{.j}) - \mathrm{Cov}(Z, Y_{.j})\mathrm{Var}(Z)^{-1}\mathrm{Cov}(Z, Y_{.j})^T\big|\Omega_{.m} = 1\big).$$

One has

$$\mathrm{Cov}(Z, Y_{.j})\mathrm{Var}(Z)^{-1}\mathrm{Cov}(Z, Y_{.j})^T =$$
$$\mathrm{Cov}((Y_{.k})_{k\in\overline{\{j\}}}, Y_{.j})\mathrm{Var}((Y_{.k})_{k\in\overline{\{j\}}})^{-1}\mathrm{Cov}((Y_{.k})_{k\in\overline{\{j\}}}, Y_{.j})^T$$

leading to

$$\mathbb{E}[\mathrm{Var}(Y_{.j}|Z)] = Q^c, \tag{39}$$

where $Q^c$ is defined in (37).

For the second term of (38), remark that **A2.** implies that

$$\mathrm{Var}(\mathbb{E}[Y_{.j}|Z]) = \mathrm{Var}(\mathbb{E}[Y_{.j}|Z, \Omega_{.m} = 1]),$$

and

$$\mathrm{Var}(\mathbb{E}[Y_{.j}|Z, \Omega_{.m} = 1]) = \mathrm{Var}\left(\mathbb{E}\left[\mathcal{B}^c_{j\to m, \mathcal{J}_{-j}[0]} + \sum_{k\in\{m\}\cup\mathcal{J}_{-j}} \mathcal{B}^c_{j\to m, \mathcal{J}_{-j}[k]}Y_{.k} + \zeta^c \middle| Z\right]\right),$$

*i.e.*

$$\mathrm{Var}(\mathbb{E}[Y_{.j}|Z, \Omega_{.m} = 1])$$
$$= \mathrm{Var}\left(\sum_{k\in\{m\}\cup\mathcal{J}_{-j}} \mathcal{B}^c_{j\to m, \mathcal{J}_{-j}[k]}Y_{.k} - \sum_{k\in\{m\}\cup\mathcal{J}_{-j}} \mathcal{B}^c_{j\to m, \mathcal{J}_{-j}[k]}\mathbb{E}[\epsilon_{.k}|Z] + \mathcal{B}^c_{j\to m, \mathcal{J}_{-j}[0]} + \mathbb{E}[\epsilon_{.j}]\right)$$

In the variance, the first term is obtained using that the variables $(Y_{.k})_{k\in\{m\}\cup\mathcal{J}_{-j}}$ are $Z-$measurable. The two last terms use that $\mathcal{B}^c_{j\to m, \mathcal{J}_{-j}[0]}$ is a constant and $\epsilon_{.j}$ is independent of $Z$. To calculate the second term, involving $\mathbb{E}[\epsilon_{.k}|Z]$, one first shows that the vector $\big((Y_{.k})_{k\in\{m\}\cup\mathcal{J}_{-j}} \quad (\epsilon_{.k})_{k\in\{m\}\cup\mathcal{J}_{-j}}\big)^T$ is Gaussian. Indeed,

- $(Y_{.k})_{k\in\{m\}\cup\mathcal{J}_{-j}}$ is a Gaussian vector, using the model (1).

- $(\epsilon_{.k})_{k\in\{m\}\cup\mathcal{J}_{-j}}$ is a Gaussian vector, because its components are independent Gaussian variables.

- for $k \neq \ell \in \{m\} \cup \mathcal{J}_{-j}$, $(WB_{k.} \quad \epsilon_{.\ell})^T$ is a Gaussian vector, because $Y_{.k} \perp\!\!\!\perp \epsilon_{.\ell}$.

- for $k \in \{m\} \cup \mathcal{J}_{-j}$, $(Y_{.k} \quad \epsilon_{.k})^T$ is a Gaussian vector, given that $Y_{.k}$ is a linear combination of $(WB_{k.} \quad \epsilon_{.k})^T$ which is Gaussian, as $WB_{k.}$ and $\epsilon_{.k}$ are independent Gaussian variables.

Thus,

$$\mathbb{E}[\epsilon_{.k}|Z] = \mathbb{E}[\epsilon_{.k}] + \mathrm{Cov}(\epsilon_{.k}, Z)\mathrm{Var}(Z)^{-1}(Z - \mathbb{E}[Z])$$
$$= \mathrm{Cov}(\epsilon_{.k}, Y_{.k})(\mathrm{Var}(Z)^{-1})_{k.}(Z - \mathbb{E}[Z]),$$

using $\mathrm{Cov}(\epsilon_{.k}, Y_{.l}) = 0$, for $k \neq l$. $\Gamma_Z = \mathrm{Var}(Z)^{-1}$ denotes the inverse of the covariance matrix of $Z$ and $(\Gamma_Z)_{k.}$ is its k-th row. It leads to

$$\mathbb{E}[\epsilon_{.k}|Z] = \sigma^2 (\Gamma_Z)_{k.}(Z - \mathbb{E}[Z]). \qquad (40)$$

given that $\mathrm{Cov}(\epsilon_{.k}, Y_{.k}) = \mathrm{Cov}(\epsilon_{.k}, WB_{k.} + \epsilon_{.k}) = \mathrm{Var}(\epsilon_{.k})$.

Therefore,

$$\mathrm{Var}(\mathbb{E}[Y_{.j}|Z, \Omega_{.m} = 1]) = \sum_{k \in \{m\} \cup \mathcal{J}_{-j}} (\mathcal{B}^c_{j \to m, \mathcal{J}_{-j}[k]})^2 \mathrm{Var}(Y_{.k})$$

$$+ \sum_{(k < \ell) \in \{m\} \cup \mathcal{J}_{-j}} 2\mathcal{B}^c_{j \to m, \mathcal{J}_{-j}[k]} \mathcal{B}^c_{j \to m, \mathcal{J}_{-j}[\ell]} \mathrm{Cov}(Y_{.k}, Y_{.\ell}) + o_{\mathrm{var}}(\sigma^2), \quad (41)$$

where

$$o_{\mathrm{var}}(\sigma^2) = -2\sigma^2 \sum_{(k,\ell) \in \{m\} \cup \mathcal{J}_{-j}} \mathcal{B}^c_{j \to m, \mathcal{J}_{-j}[k]} \mathcal{B}^c_{j \to m, \mathcal{J}_{-j}[\ell]} \sum_{\ell' \in \{m\} \cup \mathcal{J}_{-j}} (\Gamma_Z)_{\ell\ell'} \mathrm{Cov}(Y_{.k}, Y_{.\ell'})$$

$$+ \sigma^4 \sum_{k \in \{m\} \cup \mathcal{J}_{-j}} (\mathcal{B}^c_{j \to m, \mathcal{J}_{-j}[k]})^2 \left( \sum_{(\ell < \ell') \in \{m\} \cup \mathcal{J}_{-j}} (\Gamma_Z)^2_{k\ell} \mathrm{Var}(Y_{.\ell}) - 2(\Gamma_Z)_{k\ell}(\Gamma_Z)_{k\ell'} \mathrm{Cov}(Y_{.\ell}, Y_{.\ell'}) \right)$$

$$- 2\sigma^4 \sum_{(k < \ell) \in \{m\} \cup \mathcal{J}_{-j}} \mathcal{B}^c_{j \to m, \mathcal{J}_{-j}[k]} \mathcal{B}^c_{j \to m, \mathcal{J}_{-j}[\ell]} \sum_{(k',\ell') \in \{m\} \cup \mathcal{J}_{-j}} (\Gamma_Z)_{kk'}(\Gamma_Z)_{\ell\ell'} \mathrm{Cov}(Y_{.k'}, Y_{.\ell'}) \quad (42)$$

Combining (39) with (41), one get the following expression for the first line of the linear system

$$(\mathcal{B}^c_{j \to m, \mathcal{J}_{-j}[m]})^2 \mathrm{Var}(Y_{.m}) + \sum_{j' \in \mathcal{J}_{-j}} 2\mathcal{B}^c_{j \to m, \mathcal{J}_{-j}[j']} \mathcal{B}^c_{j \to m, \mathcal{J}_{-j}[m]} \mathrm{Cov}(Y_{.j'}, Y_{.m})$$

$$= \mathrm{Var}(Y_{.j}) - Q^c - (\mathcal{B}^c_{j \to m, \mathcal{J}_{-j}[\mathcal{J}_{-j}]})^T \mathrm{Var}(Y_{\mathcal{J}_{-j}}) \mathcal{B}^c_{j \to m, \mathcal{J}_{-j}[\mathcal{J}_{-j}]} - o_{\mathrm{var}}(\sigma^2) \quad (43)$$

**Deriving equations for the covariances.** Let $k$ be an element of $\mathcal{J}$, our objective is to express $\mathrm{Cov}(Y_{.m}, Y_{.k})$ from $\mathrm{Var}(Y_{.m})$, $\alpha_m$, $(\alpha_k)_{k \in \mathcal{J}}$ and $(\mathrm{Cov}(Y_{.m}, Y_{.k}))_{k \in \{m\} \cup \mathcal{J}}$.

$$\begin{aligned} \mathrm{Cov}(Y_{.m}, Y_{.k}) &= \mathbb{E}[Y_{.m}Y_{.k}] - \mathbb{E}[Y_{.m}]\mathbb{E}[Y_{.k}] \\ &= \mathbb{E}[\mathbb{E}[Y_{.m}Y_{.k}|Z]] - \mathbb{E}[Y_{.m}]\mathbb{E}[Y_{.k}] \\ &= \mathbb{E}[Y_{.m}\mathbb{E}[Y_{.k}|Z]] - \mathbb{E}[Y_{.m}]\mathbb{E}[Y_{.k}], \end{aligned} \qquad (44)$$

with $Z = (Y_{.\ell})_{\ell \in \overline{\{k\}}}$.

For the first term in (44), one has

$$\mathbb{E}[Y_{.m}\mathbb{E}[Y_{.k}|Z]] \overset{(i)}{=} \mathbb{E}[Y_{.m}\mathbb{E}[Y_{.k}|Z, \Omega_{.m} = 1]]$$

$$\overset{(ii)}{=} \mathbb{E}\left[ Y_{.m} \left( \mathcal{B}^c_{k \to m, \mathcal{J}_{-k}[0]} + \sum_{\ell \in \{m\} \cup \mathcal{J}_{-k}} \mathcal{B}^c_{k \to m, \mathcal{J}_{-k}[\ell]} Y_{.\ell} + \mathbb{E}[\zeta^c_k|Z] \right) \right]$$

$$\overset{(iii)}{=} \mathcal{B}^c_{k \to m, \mathcal{J}_{-k}[0]} \mathbb{E}[Y_{.m}] + \mathcal{B}^c_{k \to m, \mathcal{J}_{-k}[m]} \mathbb{E}[Y^2_{.m}]$$

$$+ \sum_{\ell \in \mathcal{J}_{-k}} \mathcal{B}^c_{k \to m, \mathcal{J}_{-k}[\ell]} \mathbb{E}[Y_{.m}Y_{.\ell}] + o_{\mathrm{cov},k}(\sigma^2)$$

with $\zeta^c_k = -\sum_{\ell \in \mathcal{J}_{-k}} \mathcal{B}^c_{k \to m, \mathcal{J}_{-k}[\ell]} \epsilon_{.\ell} - \mathcal{B}^c_{k \to m, \mathcal{J}_{-k}[m]} \epsilon_{.m} + \epsilon_{.k}$.

Assumption **A2.** and Definition 3 are used for (i) and (ii) respectively. For (iii), using (40), one has

$$\mathbb{E}[Y_{.m}\mathbb{E}[\zeta^c_k|Z]] = \mathbb{E}\left[ Y_{.m} \left( -\sum_{\ell \in \{m\} \cup \mathcal{J}_{-k}} \mathcal{B}^c_{k \to m, \mathcal{J}_{-k}[\ell]} \sigma^2 (\Gamma_Z)_{\ell.}(Z - \mathbb{E}[Z]) \right) \right],$$

given that $\mathbb{E}[\epsilon_{.k}|Z] = \mathbb{E}[\epsilon_{.k}] = 0$ by independence.

$$\mathbb{E}[Y_{.m}\mathbb{E}[\zeta^c_k|Z]]$$

$$= -\sigma^2 \mathbb{E}\left[ \sum_{\ell \in \{m\} \cup \mathcal{J}_{-k}} \mathcal{B}^c_{k \to m, \mathcal{J}_{-k}[\ell]} Y_{.m} \sum_{\ell' \{m\} \cup \in \mathcal{J}_{-k}} (\Gamma_Z)_{\ell\ell'}(Y_{.\ell'} - \mathbb{E}[Y_{.\ell'}]) \right].$$

In addition,

$$\mathbb{E}\left[\sum_{\ell\in\{m\}\cup\mathcal{J}_{-k}}\mathcal{B}^c_{k\to m,\mathcal{J}_{-k}[\ell]}Y_{.m}\sum_{\ell'\in\{m\}\cup\mathcal{J}_{-k}}(\Gamma_Z)_{\ell\ell'}(Y_{.\ell'}-\mathbb{E}[Y_{.\ell'}])\right]$$

$$=\sum_{\ell\in\{m\}\cup\mathcal{J}_{-k}}\sum_{\ell'\in\{m\}\cup\mathcal{J}_{-k}}(\Gamma_Z)_{\ell\ell'}\mathcal{B}^c_{k\to m,\mathcal{J}_{-k}[\ell]}\left(\mathrm{Cov}\left(Y_{.m},Y_{.\ell'}\right)+\mathbb{E}[Y_{.m}]\mathbb{E}[(Y_{.\ell'}-\mathbb{E}[Y_{.\ell'}])]\right)$$

$$=\sum_{\ell\in\{m\}\cup\mathcal{J}_{-k}}\sum_{\ell'\in\{m\}\cup\mathcal{J}_{-k}}(\Gamma_Z)_{\ell\ell'}\mathcal{B}^c_{k\to m,\mathcal{J}_{-k}[\ell]}\mathrm{Cov}\left(Y_{.m},Y_{.\ell'}\right)$$

It implies that, in (iii),

$$o_{\mathrm{cov},k}(\sigma^2)=-\sigma^2\sum_{\ell\in\{m\}\cup\mathcal{J}_{-k}}\sum_{\ell'\in\{m\}\cup\mathcal{J}_{-k}}(\Gamma_Z)_{\ell\ell'}\mathcal{B}^c_{k\to m,\mathcal{J}_{-k}[\ell]}\mathrm{Cov}\left(Y_{.m},Y_{.\ell'}\right)\qquad(45)$$

Equation (44) leads thus to

$$\mathrm{Cov}(Y_{.m},Y_{.k})=\mathcal{B}^c_{k\to m,\mathcal{J}_{-k}[0]}\mathbb{E}[Y_{.m}]+\mathcal{B}^c_{k\to m,\mathcal{J}_{-k}[m]}(\mathrm{Var}(Y_{.m})+\mathbb{E}[Y_{.m}]^2)$$
$$+\sum_{\ell\in\mathcal{J}_{-k}}\mathcal{B}^c_{k\to m,\mathcal{J}_{-k}[\ell]}(\mathrm{Cov}(Y_{.m},Y_{.\ell})+\mathbb{E}[Y_{.m}]\mathbb{E}[Y_{.\ell}])-\mathbb{E}[Y_{.m}]\mathbb{E}[Y_{.k}]+o_{\mathrm{cov},k}(\sigma^2),\quad(46)$$

which can be rewritten as

$$\mathrm{Cov}(Y_{.m},Y_{.k})-\mathcal{B}^c_{k\to m,\mathcal{J}_{-k}[m]}\mathrm{Var}(Y_{.m})-\sum_{\ell\in\mathcal{J}_{-k}}\mathcal{B}^c_{k\to m,\mathcal{J}_{-k}[\ell]}\mathrm{Cov}(Y_{.m},Y_{.\ell})$$

$$=((\mathcal{B}^c_{k\to m,\mathcal{J}_{-k}})^T\begin{pmatrix}1&\mathbb{E}[Y_{.m}]&(\mathbb{E}[Y_{.\ell}])_{\ell\in\mathcal{J}_{-k}}\end{pmatrix}^T-\mathbb{E}[Y_{.k}])\mathbb{E}[Y_{.m}]+o_{\mathrm{cov},k}(\sigma^2),\quad(47)$$

Combining Equations (43) and (47) forms the desired matrix system (36).

From these formulae for $(\mathrm{Var}(Y_{.m})\quad\mathrm{Cov}(Y_{.m},(Y_{.k})_{k\in\mathcal{J}}))^T$, assuming that $M_j$ is invertible and that $\sigma^2$ tends to zero, one get their estimators $\left(\widehat{\mathrm{Var}}(Y_{.m})\quad\widehat{\mathrm{Cov}}(Y_{.m},(Y_{.k})_{k\in\mathcal{J}})\right)^T$ defined in (10).

As for the consistency, $\hat{\alpha}_m$ is a consistent estimator for $\alpha_m$ by using Proposition 4. The estimators in (10) are consistent, under Assumption **A3.** and **A4.**.  $\square$

## B.4 Proof of Proposition 8

For deriving the covariance between a MNAR variable and a MNAR or not pivot variable, we assume the following

**A5.** $\forall m\in\mathcal{M}, \forall\ell\in\bar{\mathcal{J}}_{-m}$, for all set $\mathcal{H}\subset\mathcal{J}_{-j}$ such that $|\mathcal{H}|=r-2$, $(B_{.m}\quad B_{.\ell}\quad(B_{.j'})_{j'\in\mathcal{H}})$ is invertible,

**A6.** $\forall k\in\bar{\mathcal{J}}\backslash\mathcal{M}, \forall j\in\mathcal{J},\ Y_{.j}\perp\!\!\!\perp\Omega_{.k}|(Y_{.\ell})_{\ell\in\overline{\{j\}}}$.

**A7.** $\forall k,\ell\in\bar{\mathcal{J}},\quad k\neq l,\ \Omega_{.k}\perp\!\!\!\perp\Omega_{.\ell}|Y$

**A8.** $\forall j\in\mathcal{J}, \forall m\in\mathcal{M}, \forall\ell\in\bar{\mathcal{J}}_{-m}$, for all set $\mathcal{H}\subset\mathcal{J}_{-j}$ such that $|\mathcal{H}|=r-2$, the complete-case coefficients $\mathcal{B}^c_{j\to m,\ell,\mathcal{H}[0]}$ and $\mathcal{B}^c_{j\to m,\ell,\mathcal{H}[k]}, k\neq j, k\in\{m,\ell\}\cup\mathcal{H}$ can be consistently estimated. (Here, note that the complete case is when $\Omega_{.m}=1$ and $\Omega_{.\ell}=1$.)

**A9.** For the variables neither MNAR nor pivot, their means $(\alpha_k)_{k\in\bar{\mathcal{J}}\backslash\mathcal{M}}$, variances $(\mathrm{Var}(Y_{.k}))_{k\in\bar{\mathcal{J}}\backslash\mathcal{M}}$ and covariances $(\mathrm{Cov}(Y_{.k},Y_{.k'}))_{k\neq k'\in\bar{\mathcal{J}}\backslash\mathcal{M}}$ can be consistently estimated. The covariances between these variables and the pivot variables $(\mathrm{Cov}(Y_{.j},Y_{.k}))_{j\in\mathcal{J},k\in\bar{\mathcal{J}}\backslash\mathcal{M}}$ are also consistent.

**Proposition 8** (Covariance between a MNAR variable and a MNAR or not pivot variable)**.** *Consider the PPCA model (1). Under Assumptions **A2.**, **A5.**, **A6.** and **A7.**, an estimator of the covariance between a MNAR variable $Y_{.m}$, for $m\in\mathcal{M}$, and a variable $Y_{.\ell}$, for $\ell\in\bar{\mathcal{J}}\backslash\{m\}$, can be constructed*

*as follows: choose $j \in \mathcal{J}$ and $r - 2$ variable indexes in $\mathcal{J}_{-j}$ and compute:*

$$\widehat{\mathrm{Cov}}(Y_{.m}, Y_{.\ell}) = \frac{1}{\hat{K}} \widehat{\mathrm{Var}}(Y_{.j}) - \hat{q}^c - \sum_{k \in \{m,\ell\} \cup \mathcal{H}} (\hat{\mathcal{B}}^c_{j \to m,\ell,\mathcal{H}[k]})^2 \widehat{\mathrm{Var}}(Y_{.k})$$

$$- \sum_{k < k', k \in \{m,\ell\} \cup \mathcal{H}, k' \in \mathcal{H}} 2\hat{\mathcal{B}}^c_{j \to m,\ell,\mathcal{H}[k]} \hat{\mathcal{B}}^c_{j \to m,\ell,\mathcal{H}[k']} \widehat{\mathrm{Cov}}(Y_{.k}, Y_{.k'}), \quad (48)$$

*assuming that $\sigma^2$ tends to zero and with $\hat{K} = 2\hat{\mathcal{B}}^c_{j \to m,\ell,\mathcal{H}[m]} \hat{\mathcal{B}}^c_{j \to m,\ell,\mathcal{H}[\ell]}$ and*

$$\hat{q}^c = \left( \widehat{\mathrm{Var}}(Y_{.j}) \big| \Omega_{.m} = 1, \Omega_{.\ell} = 1 \right)$$
$$- \left( \widehat{\mathrm{Cov}}((Y_{.k})_{k \in \overline{\{j\}}}, Y_{.j}) \widehat{\mathrm{Var}}((Y_{.k})_{k \in \overline{\{j\}}})^{-1} \widehat{\mathrm{Cov}}((Y_{.k})_{k \in \overline{\{j\}}}, Y_{.j})^T \big| \Omega_{.m} = 1, \Omega_{.\ell} = 1 \right),$$

*given that $K$ estimated by $\hat{K}$ is non zero.*

*Under the additional Assumptions **A3., A8.** and **A9.**. this estimator given in (48) is consistent.*

*Proof.* Let $\mathcal{H}$ be the set of the $r - 2$ variable indexes. One has $\mathcal{H} \subset \mathcal{J}_{-j}$. We use the same strategy as the proof for Proposition 5 (paragraph for deriving an equation for the variance).

To derive a formula for $\mathrm{Cov}(Y_{.m}, Y_{.\ell})$ with $m \in \mathcal{M}$ and $\ell \in \bar{\mathcal{J}}_{-m}$, the idea is to express $\mathrm{Var}(Y_{.j})$ from $(\mathrm{Var}(Y_{.k}))_{k \in \{m,\ell\} \cup \mathcal{H}}$ and $(\mathrm{Cov}(Y_{.k}, Y_{.k'}))_{k \neq k' \in \{m,\ell\} \cup \mathcal{H}}$.

The law of total variance reads as

$$\mathrm{Var}(Y_{.j}) = \mathbb{E}[\mathrm{Var}(Y_{.j}|Z)] + \mathrm{Var}(\mathbb{E}[Y_{.j}|Z]), \quad (49)$$

with $Z = (Y_{.k})_{k \in \overline{\{j\}}}$.

For the first term in (49), one uses

$$Y_{.j} \perp\!\!\!\perp \Omega_{.m}, \Omega_{.\ell} | Z.$$

If $Y_{.m}$ and $Y_{.\ell}$ are both MNAR variables, this conditional independence is obtained using Assumption **A2.** and **A7.**. Otherwise, if $Y_{.\ell}$ is not a MNAR variable, Assumption **A6.** and **A7.** lead to the desired result. It implies

$$\mathrm{Var}(Y_{.j}|Z) = \mathrm{Var}(Y_{.j}|Z, \Omega_{.m} = 1, \Omega_{.\ell} = 1).$$

The conditional variance for a Gaussian vector gives

$$\mathrm{Var}(Y_{.j}|Z) = \mathrm{Var}(Y_{.j}) - \mathrm{Cov}(Z, Y_{.j}) \mathrm{Var}(Z)^{-1} \mathrm{Cov}(Z, Y_{.j})^T,$$

implying that

$$\mathrm{Var}(Y_{.j}|Z, \Omega_{.m} = 1, \Omega_{.\ell} = 1)$$
$$= \left( \mathrm{Var}(Y_{.j}) - \mathrm{Cov}(Z, Y_{.j}) \mathrm{Var}(Z)^{-1} \mathrm{Cov}(Z, Y_{.j})^T \big| \Omega_{.m} = 1, \Omega_{.\ell} = 1 \right)$$

and then, as deterministic quantity,

$$\mathbb{E}[\mathrm{Var}(Y_{.j}|Z)] = q^c \quad (50)$$

with

$$q^c = \left( \mathrm{Var}(Y_{.j}) \big| \Omega_{.m} = 1, \Omega_{.\ell} = 1 \right)$$
$$- \left( \mathrm{Cov}((Y_{.k})_{k \in \overline{\{j\}}}, Y_{.j}) \mathrm{Var}((Y_{.k})_{k \in \overline{\{j\}}})^{-1} \mathrm{Cov}((Y_{.k})_{k \in \overline{\{j\}}}, Y_{.j})^T \big| \Omega_{.m} = 1, \Omega_{.\ell} = 1 \right).$$

For the second term of (38), remark that **A2., A6.** and **A7.** implies that

$$\mathrm{Var}(\mathbb{E}[Y_{.j}|Z]) = \mathrm{Var}(\mathbb{E}[Y_{.j}|Z, \Omega_{.m} = 1, \Omega_{.\ell} = 1]),$$

and

$$\mathrm{Var}(\mathbb{E}[Y_{.j}|Z, \Omega_{.m} = 1, \Omega_{.\ell} = 1])$$

$$= \mathrm{Var}\left( \mathbb{E}\left[ \mathcal{B}^c_{j \to m,\ell,\mathcal{H}[0]} + \sum_{k \in \{m,\ell\} \cup \mathcal{H}} \mathcal{B}^c_{j \to m,\ell,\mathcal{H}[k]} Y_{.k} + \zeta^c_j \Big| Z \right] \right),$$

*i.e.*

$$\text{Var}(\mathbb{E}[Y_{.j}|Z, \Omega_{.m}=1, \Omega_{.\ell}=1])$$

$$= \text{Var}\left( \sum_{k\in\{m,\ell\}\cup\mathcal{H}} \mathcal{B}^c_{j\to m,\ell,\mathcal{H}[k]}Y_{.k} - \sum_{k\in\{m,\ell\}\cup\mathcal{H}} \mathcal{B}^c_{j\to m,\ell,\mathcal{H}[k]}\mathbb{E}[\epsilon_{.k}|Z] + \mathcal{B}^c_{j\to m,\ell,\mathcal{H}[0]} + \mathbb{E}[\epsilon_{.j}] \right)$$

One uses the same reasoning as in the proof of Proposition 5 (paragraph for deriving an equation for the variance) to get

$$\text{Var}(\mathbb{E}[Y_{.j}|Z, \Omega_{.m}=1, \Omega_{.\ell}=1]) = \sum_{k\in\{m,\ell\}\cup\mathcal{H}} (\mathcal{B}^c_{j\to m,\ell,\mathcal{H}[k]})^2\text{Var}(Y_{.k})$$

$$+ \sum_{k<k'\in\{m,\ell\}\cup\mathcal{H}} 2\mathcal{B}^c_{j\to m,\ell,\mathcal{H}[k]}\mathcal{B}^c_{j\to m,\ell,\mathcal{H}[k']}\text{Cov}(Y_{.k}, Y_{.k'}) + o_{\text{covmiss}}(\sigma^2), \quad (51)$$

where

$$o_{\text{covmiss}}(\sigma^2) = -2\sigma^2 \sum_{(k,k')\in\{m,\ell\}\cup\mathcal{H}} \mathcal{B}^c_{j\to m,\ell,\mathcal{H}[k]}\mathcal{B}^c_{j\to m,\ell,\mathcal{H}[k']} \sum_{\ell'\in\{m,\ell\}\cup\mathcal{H}} (\Gamma_Z)_{k'\ell'}\text{Cov}(Y_{.k}, Y_{.\ell'})$$

$$+ \sigma^4 \sum_{k\in\{m,\ell\}\cup\mathcal{H}} (\mathcal{B}^c_{j\to m,\ell,\mathcal{H}[k]})^2 \left( \sum_{(k'<\ell')\in\{m,\ell\}\cup\mathcal{H}} (\Gamma_Z)^2_{kk'}\text{Var}(Y_{.k'}) - 2(\Gamma_Z)_{kk'}(\Gamma_Z)_{k\ell'}\text{Cov}(Y_{.k'}, Y_{.\ell'}) \right)$$

$$- 2\sigma^4 \sum_{(k<k')\in\{m,\ell\}\cup\mathcal{H}} \mathcal{B}^c_{j\to m,\ell,\mathcal{H}[k]}\mathcal{B}^c_{j\to m,\ell,\mathcal{H}[k']} \sum_{(k'',\ell')\in\{m,\ell\}\cup\mathcal{H}} (\Gamma_Z)_{kk''}(\Gamma_Z)_{k'\ell'}\text{Cov}(Y_{.k''}, Y_{.\ell'}) \quad (52)$$

Combining (49), (50) and (51), one get the following formula for $\text{Cov}(Y_{.m}, Y_{.\ell})$,

$$2\mathcal{B}^c_{j\to m,\ell,\mathcal{H}[m]}\mathcal{B}^c_{j\to m,\mathcal{H}[\ell]}\text{Cov}(Y_{.m}, Y_{.\ell}) = \text{Var}(Y_{.j}) - q^c - \sum_{k\in\{m,\ell\}\cup\mathcal{H}} (\mathcal{B}^c_{j\to m,\ell,\mathcal{H}[k]})^2\text{Var}(Y_{.k})$$

$$- \sum_{k<k',k\in\{m,\ell\}\cup\mathcal{H},k'\in\mathcal{H}} 2\mathcal{B}^c_{j\to m,\ell,\mathcal{H}[k]}\mathcal{B}^c_{j\to m,\ell,\mathcal{H}[k']}\text{Cov}(Y_{.k}, Y_{.k'}) - o_{\text{covmiss}}(\sigma^2)$$

An estimator of $\text{Cov}(Y_{.m}, Y_{.\ell})$ is then derived as in (48), given that $\sigma^2$ tends to zero and $K = \mathcal{B}^c_{j\to m,\ell,\mathcal{H}[m]}\mathcal{B}^c_{j\to m,\ell,\mathcal{H}[\ell]}$ is non zero.

We use the consistent estimators defined in Proposition 5 for estimating $\text{Var}(Y_{.m})$ and $\text{Cov}(Y_{.m}, Y_{.k})_{k\in\mathcal{H}}$. If $Y_{.\ell}$ is also a MNAR variable, Proposition 5 is applied for estimating $\text{Var}(Y_{.\ell})$ and $\text{Cov}(Y_{.\ell}, Y_{.k})_{k\in\mathcal{H}}$. Otherwise, if $Y_{.\ell}$ is not a MNAR variable, we use **A9.**.

Eventually, **A3.** and **A8.** lead to the consistency of $\widehat{\text{Cov}}(Y_{.m}, Y_{.\ell})$. $\qquad\square$

## B.5 Extension to more general mechanisms for the not MNAR variables

The results of Proposition 4, 5 and 8 can be extended to a more general setting than the one presented in Section 2. The pivot variables may be assumed to be MCAR (or observed). The variables which are neither MNAR nor pivot may be observed or satisfying

$$\forall \ell \in \bar{\mathcal{J}}\backslash\mathcal{M}, \forall i \in \{1,\dots,n\}, \quad \mathbb{P}(\Omega_{i\ell}=1|Y_{i.}) = \mathbb{P}(\Omega_{i\ell}=1|(Y_{ik})_{k\in\bar{\mathcal{J}}\backslash\{\ell\}\cup\mathcal{M}}), \quad (53)$$

*i.e.* they are MCAR or MAR but their missing-data mechanisms may not depend on the pivot variables.

The proofs are similar and not presented here for the sake of brevity.

Note that the main difference is that the complete case has to be extended. For instance, for $j \in \mathcal{J}$ and $k \in \mathcal{J}_{-j}$, the coefficients standing respectively for the intercept and the effects of $Y_{.j}$ on $(Y_{.m}, (Y_{.j'})_{j'\in\mathcal{J}_{-j}})$ in the complete case, *i.e.* when $\Omega_{.m}=1, (\Omega_j=1)_{j\in\mathcal{J}}$ are in this general setting defined as follows

$$(Y_{.j})_{|\Omega_{.m}=1,(\Omega_j=1)_{j\in\mathcal{J}}} := \mathcal{B}^c_{j\to m,\mathcal{J}_{-j}[0]} + \sum_{j'\in\mathcal{J}_{-j}} \mathcal{B}^c_{j\to m,\mathcal{J}_{-j}[j']}Y_{.j'} + \mathcal{B}^c_{j\to m,\mathcal{J}_{-j}[m]}Y_{.m} + \zeta^c,$$

with $\zeta^c = -\sum_{j'\in\mathcal{J}_{-j}} \mathcal{B}^c_{j\to m,\mathcal{J}_{-j}[j']}\epsilon_{.j'} - \mathcal{B}^c_{j\to m,\mathcal{J}_{-j}[m]}\epsilon_{.m} + \epsilon_{.j}$.

# C  Other numerical experiments

**Robustness to noise.**  Considering the same setting as in Section 4.1 ($n = 1000$, $p = 10$, $r = 2$ and seven self-masked MNAR variables), the methods are tried for different noise levels $\sigma^2 \in \{0.1, 0.3, 0.5, 0.7, 1\}$. The results are presented for one missing variable and for all the other ones, the results are similar. In Figure 7, Algorithm 1 is the only method that does not give a biased estimate of the mean and the variance regardless of the noise level. In Figure 8, despite a larger bias in the estimation of the covariance between a missing variable and a pivot one as the noise level increases, Algorithm 1 outperforms all the other methods, regarding the estimation of the covariance between two missing variables. Note that the formula for the estimate of the covariance between two missing variables relies on the one for the estimate of the variance, but both differ from the one used for the covariance estimation between a missing variable and a pivot one. As expected, in Figure 9, estimation deteriorates as the data gets noisier and then the loading matrix estimation and the imputation error get closer to the results of mean imputation. In term of imputation error, the proposed method yet remains competitive in regards of the approaches (ii) and (iii). Overall, when the noise level increases, the exogeneity will be worse and that ignoring it in practice can be made to the detriment of performance.

Figure 7:  Mean estimation (left graphic) and variance estimation (right graphic) of one missing variable for different values of the level of noise when $r = 2$, $n = 1000$, $p = 10$ and seven variables are MNAR. True values to be estimated are indicated by red lines.

Figure 8:  Covariance estimation beetween a missing variable and a pivot one (left graphic) and two missing variables (right graphic) for different values of the level of noise when $r = 2$, $n = 1000$, $p = 10$ and seven variables are MNAR. True values to be estimated are indicated by red lines.

Figure 9:  RV coefficients for the loading matrix (left graphic) and imputation error (right graphic) for different values of the level of noise when $r = 2$, $n = 1000$, $p = 10$ and seven variables are MNAR.

**Varying the percentage of missing values.** Considering the same setting as in Section 4.1 ($n = 1000$, $p = 10$, $r = 2$, $\sigma = 0.1$ and seven self-masked MNAR variables), the methods are tried for different percentages of missing values (10%, 30%, 50%). The results are presented in Figure 10. As expected, all the methods deteriorate with an increasing percentage of missing values but our method is stable.

Figure 10: Mean estimation (left graphic), variance estimation (middle graphic) and imputation error (right graphic) for different percentages of missing values when $r = 2$, $n = 1000$, $p = 10$ and seven variables are MNAR.

**Misspecification to the rank.** The misspecification to the parameter $r$ has been evaluated: under a model generated with $r = 3$ latent variables ($n = 1000$, $p = 20$, $\sigma = 0.8$ and ten MNAR self-masked variables), the rank is either underestimated, well estimated or overestimated by giving to Algorithm 1 the information that $r = 2$, $r = 3$ or $r = 4$. Both estimation of the loading matrix and imputation error are shown in Figure 11. The results for an underestimated ($r = 2$) or overestimated ($r = 4$) rank are comparable to the case where the accurate rank is considered instead ($r = 3$), showing a stability of Algorithm 1 to rank misspecification.

Figure 11: RV coefficients for the loading matrix (left) and imputation error (right) when $r = 3$, $n = 1000$, $p = 20$ and ten variables are MNAR for different cases where the rank is either underestimated, well estimated or overestimated.

**General MNAR mechanism.** We consider the setting $n = 1000$, $p = 20$, $r = 3$ and $\sigma = 0.8$. Here, missing values are introduced on ten variables $(Y_{.k})_{k \in [1:10]}$ using a more general MNAR mechanism (see (3)) than the self-masked one. In particular, the MNAR mechanism we consider is defined as follows,

$$\forall m \in [1:10], \forall i \in \{1, \ldots, n\}, \; \mathbb{P}(\Omega_{im} = 1 | Y_{i.}) = \mathbb{P}(\Omega_{im} = 1 | Y_{im}, Y_{ik}, Y_{i\ell}), \qquad (54)$$

where $k$ and $\ell$ are indexes of MNAR variables randomly chosen such that $k \neq \ell \in [1:10] \backslash \{m\}$. In Figure 12, Algorithm 1 provides the best estimators of the mean and the variance (in term of bias) and the smallest imputation error.

**Higher dimension and variation of the rank.** The performance of the different methods for higher dimension is assessed. A data matrix of size $n = 1000$ and $p = 50$ is generated from two

Figure 12: Mean estimation (left), variance estimation (middle) of one missing variable and imputation error (right) when $r = 3$, $n = 1000$, $p = 20$ and ten variables are MNAR as in (54). True values are indicated in red lines.

Figure 13: Mean estimation (left) and variance estimation (right) of one missing variable when $r = 2$, $n = 1000$, $p = 50$ and twenty variables are MNAR. True values to be estimated are indicated by red lines.

latent variables ($r = 2$) and with a noise level $\sigma = 1$. Missing values are introduced on twenty variables according to a self-masked MNAR mechanism, leading to 20% of missing values in total. Without loss of generality, the results are presented for one missing variable. Method (iv) has been discarded, as its computational time is too high for this setting.

In Figure 13, as for the estimated mean and variance, Methods (i), (ii) and (iii) suffer from a large bias, while Algorithm 1 gives unbiased estimators. The same comment can be done for the estimation of the covariance between two missing values in Figure 14. As for the covariance estimation between a missing variable and a pivot one Figure 14, Algorithm 1 suffers from a variability, which can be due to the fact that in this higher dimension setting, not all the possible combinations of pivot variables are considered. Indeed, instead of taking the set of pivot variables of all the not MNAR variables *i.e.* $\mathcal{J} = \overline{\mathcal{M}}$, we choose $\mathcal{J} \subset \overline{\mathcal{M}}$ such that $|\mathcal{J}| = 10$. For the mean, 270 combinations of the pivot variables are aggregated over 870 possible combinations if $\mathcal{J} = \overline{\mathcal{M}}$.

Figure 14: Covariance estimation beetween two missing variable (left) and a missing variable and a pivot one (right) when $r = 2$, $n = 200$, $p = 10$ and seven variables are MNAR. True values to be estimated are indicated by red lines.

Figure 15: RV coefficients for the loading matrix (left) and imputation error (right) when $r = 2$, $n = 1000$, $p = 50$ and twenty variables are MNAR.

Despite this dispersed estimator of the covariance between a MNAR variable and a pivot one, Algorithm 1 gives in Figure 15 a high RV coefficient, by improving Methods (i), (iii) and (ii). Concerning the imputation performance, Algorithm 1 strongly improves Methods (ii) and (iii).

For the same dimension setting ($n = 1000$, $p = 50$) and the same noise level ($\sigma = 1$), we vary the rank to $r = 5$. Similarly as before, missing values are introduced on twenty variables according to a self-masked MNAR mechanism, leading to 20% of missing values in total. In Figure 16, for the mean and the variable estimations, Algorithm 1 gives unbiased estimators. In Figure 17, the covariance between a missing variable and a pivot estimated by Algorithm 1 is biased but still less than the other methods. In addition, the covariance between two missing variables is unbiased but suffers from a high variability. Note that once again we have chosen $\mathcal{J} \subset \mathcal{M}$ such that $|\mathcal{J}| = 10$. For the mean, 1260 combinations of the pivot variables are aggregated over 712530 possible combinations if $\mathcal{J} = \overline{\mathcal{M}}$. In Figure 18, despite such results for the covariance estimators, Algorithm 1 gives a similar RV coefficient than Methods (ii) and (iii) but strongly improves all the methods in term of imputation error.

Figure 16: Mean estimation (left) and variance estimation (right) of one missing variable when $r = 5$, $n = 1000$, $p = 50$ and twenty variables are MNAR. True values to be estimated are indicated by red lines.

Figure 17: Covariance estimation beetween two missing variable (left) and a missing variable and a pivot one (right) when $r = 5$, $n = 1000$, $p = 50$ and twenty variables are MNAR. True values to be estimated are indicated by red lines.

Figure 18: RV coefficients for the loading matrix (left) and imputation error (right) when $r = 5$, $n = 1000$, $p = 50$ and twenty variables are MNAR.

**Efficiency of the *aggregation* approach in the selection of the pivot variables.** As described in Section 3.4, Algorithm 1 requires the selection of $r$ pivot variables (considered M(C)AR) on which the regressions will be performed. To reduce the error committed by the selection pivot variables, we propose to select a bigger set of pivot variables (with a cardinal superior to $r$) and the final estimator will be computed with the median of the estimators over all possible combinations of $r$ pivot variables (this is called the *aggregation* approach). In Figure 19, we consider the same setting as in Section 4.1 ($n = 1000$, $p = 10$, $r = 2$ and seven self-masked MNAR variables) and we perform Algorithm 1 by using the *aggregation* (`MNARagg`) method or not (`MNARnoagg`). By discarding outliers, this *aggregation* approach is more robust than selecting only $r$ pivot variables.

Figure 19: Mean (left) and variance (middle left) estimations of $Y_{.1}$ and covariances estimations of $\mathrm{Cov}(Y_{.1}, Y_{.2})$ (between two missing variables) (middle right) and of $\mathrm{Cov}(Y_{.1}, Y_{.8})$ (between one missing variable and one pivot variable) (right). True values are indicated in red lines.

## D  Computation time

Table 1 gathers computation times of the different methods, for both settings considered in Sections 4 and C.

| Method | $r = 2, p = 10, n = 1000$ 35% MNAR values in 7 variables | $r = 5, p = 50, n = 1000$ 20% MNAR values in 20 variables |
|---|---|---|
| MNAR algebraic | 0,1 s | 11 min 48 s (1260 aggregations) |
| SoftMAR | 5,5 s | 28 s |
| EMMAR | 50,8 s | 2 min 9 s |
| Param | 5 h 15 min | not evaluated |

Table 1: Computation time for simulations in Sections 4 and Appendix C. The process time is obtained for a computer with a processor Intel Core i5 of 2,3 GHz.

# E  Additional information on the TraumaBase® dataset

## E.1  Description of the variables

A description of the variables which are used in Section 4.2 is given. The indications given in parentheses ph (pre-hospital) and h (hospital) mean that the measures have been taken before the arrival at the hospital and at the hospital.

- *SBP.ph*, *DBP.ph*, *HR.ph*: systolic and diastolic arterial pressure and heart rate during pre-hospital phase. (ph)

- *HemoCue.init*: prehospital capillary hemoglobin concentration. (ph)

- *SpO2.min*: peripheral oxygen saturation, measured by pulse oxymetry, to estimate oxygen content in the blood. (ph)

- *Cristalloid.volume*: total amount of prehospital administered cristalloid fluid resuscitation (volume expansion). (ph)

- *Shock.index.ph*: ratio of heart rate and systolic arterial pressure during pre-hospital phase. (ph)

- *Delta.shock.index*: Difference of shock index between arrival at the hospital and arrival on the scene. (h)

- *Delta.hemoCue*: Difference of hemoglobin level between arrival at the hospital and arrival on the scene. (h)

The percentage of missing values in each variable is given in Figure 20.

Figure 20: Percentage of missing values in each variable for the TraumaBase data.

## E.2  Supervised learning task

To predict the administration or not of the tranexomic acid (binary variable), we impute explanatory variables before proceeding to the classification task. In Table 2, Algorithm 1 gives the smallest prediction error.

| | |
|---|---|
| MNAR | 5.06% |
| EMMAR | 5.82% |
| SoftMAR | 5.45% |
| MNARparam | 5.39% |
| Mean | 5.27% |

Table 2: Mean of prediction error over 10 repetitions.

# F   Graphical approach

## F.1   Preliminaries

Lemmas of Mohan et al. [18] are used to construct some estimators of the mean, variance and covariances for a MNAR variable based on a graphical approach.

**Lemma 9** (Lemma 2 [18]). *Let us consider the m-graph $G$. The coefficient of the linear regression of $Y_{\cdot j}$ on $Y_{\cdot k}, k \neq j$, denoted as $\beta_{j \to k, k \neq j}$ is recoverable (i.e. they are consistent in the complete-case analysis) if $Y_{\cdot j} \perp\!\!\!\perp \Omega | Y_{\cdot k}, k \neq j$ and one has*

$$\beta_{j \to k, k \neq j} = \beta_{j \to k, k \neq j}^c.$$

**Lemma 10** (Lemma 1). *[18]](Graphical approach for computing the covariance) Let $G$ be a m-graph with $k$ unblocked paths $p_1, \ldots, p_k$ between two variables $Y_{\cdot \tau}$ and $Y_{\cdot \delta}$. Let $A_{p_i}$ be the ancestor of all nodes on path $p_i$. Let the number of nodes on $p_i$ be $n_{p_i}$. One can derive that*

$$\mathrm{Cov}(Y_{\cdot \tau}, Y_{\cdot \delta}) = \sum_{i=1}^{k} \mathrm{Var}(A_{p_i}) \prod_{j=1}^{n_{p_i}-1} \alpha_j^{p_i},$$

*where $\prod_{j=1}^{n_{p_i}-1} \alpha_j^{p_i}$ is the product of all causal parameters on path $p_i$.*

In addition, let us recall the basic formula,

$$\beta_{Y \to X} = \frac{\mathrm{Cov}(X, Y)}{\mathrm{Var}(X)}, \tag{55}$$

where $Y$ and $X$ are two variables of a linear model.

## F.2   Estimation of the mean, variance and covariances of the MNAR variables

The graphical approach to construct an estimator of $\alpha_1$ is based on the transformation illustrated in Figure 1 of the graphical model of PPCA as structural causal graphs, whose context is introduced in [21]. This latter framework allows to directly apply the results of Mohan et al. [18] who consider the associated (linear) structural causal equations under the exogeneity assumption with MNAR missing values for one variable.

For the sake of brevity, the results are presented for the toy example in Section 3.1 where $p = 3$, $r = 2$, $Y_{\cdot 1}$ is self-masked MNAR and the other variables are observed.

Then, one can associate to Figure 1 (bottom right graph) the structural equation model detailled in the following lemma.

**Lemma 11.** *Assuming $\mathbb{E}[\epsilon_{\cdot 2} | Y_{\cdot 1}, Y_{\cdot 3}] = 0$, the structural equation model associated with the bottom right graph in Figure 1 is*

$$Y_{\cdot 2} = \beta_{2 \to 1, 3[0]} + \beta_{2 \to 1, 3[1]} Y_{\cdot 1} + \beta_{2 \to 1, 3[3]} Y_{\cdot 3} + \epsilon_{\cdot 2}, \tag{56}$$

*where $\beta_{2 \to 1, 3[0]}$, $\beta_{2 \to 1, 3[1]}$ and $\beta_{2 \to 1, 3[3]}$ are the intercept and the coefficients of the linear regression of $Y_{\cdot 2}$ on $Y_{\cdot 1}$ and $Y_{\cdot 3}$.*

Using Equation (56) and Lemma 9, we apply the results of Mohan et al. [18] to get an estimator for the mean of the MNAR variable.

**Proposition 12** (Mean estimator for the graphical approach). *Under Equation (56), assuming **A1.** and $\beta_{2 \to 1, 3[1]}^c \neq 0$, one can construct an estimator of the mean $\alpha_1$ of the MNAR variable $Y_{\cdot 1}$ as follows*

$$\hat{\alpha}_1 := \frac{\hat{\alpha}_2 - \hat{\beta}_{2 \to 1, 3[0]}^c - \hat{\beta}_{2 \to 1, 3[3]}^c \hat{\alpha}_3}{\hat{\beta}_{2 \to 1, 3[1]}^c}, \tag{57}$$

*where $\hat{\beta}_{2 \to 1, 3[0]}^c$, $\hat{\beta}_{2 \to 1, 3[1]}^c$ and $\hat{\beta}_{2 \to 1, 3[3]}^c$ denote some estimators of $\beta_{2 \to 1, 3[0]}^c$, $\beta_{2 \to 1, 3[1]}^c$ and $\beta_{2 \to 1, 3[3]}^c$ given in Lemma 11. This estimator is consistent under additional Assumption **A4.**.*

*Proof.* To derive some estimator of the mean, we want to obtain the following formula

$$\alpha_1 = \frac{\alpha_2 - \beta^c_{2\to1,3[0]} - \beta^c_{2\to1,3[3]}\alpha_3}{\beta^c_{2\to1,3[1]}}.\tag{58}$$

Indeed, one has:

$$
\begin{aligned}
\mathbb{E}[Y_{.2}] &= \mathbb{E}[\mathbb{E}[Y_{.2}|Y_{.1},Y_{.3}] \\
&= \mathbb{E}[\mathbb{E}[Y_{.2}|Y_{.1},Y_{.3},\Omega_{.1}=1]] && \text{(by using \textbf{A1.})} \\
&= \mathbb{E}[\mathbb{E}[\beta^c_{2\to1,3[0]} + \beta^c_{2\to1,3[1]}Y_{.1} + \beta^c_{2\to3,1[3]}Y_{.3} + \epsilon_{.2}|Y_{.1},Y_{.3}]] \\
&= \beta^c_{2\to1,3[0]} + \beta^c_{2\to1,3[1]}\mathbb{E}[Y_{.1}] + \beta^c_{2\to3,1[3]}\mathbb{E}[Y_{.3}],
\end{aligned}
$$

which leads to the desired Equation (58), provided that $\beta^c_{2\to1,3[1]} \neq 0$. A natural estimator fo $\alpha_1$ is then given by (57). It is consistent given that all the quantities involved are consistent, by using **A4.** (for the consistency of $\hat{\alpha}_2$ and $\hat{\alpha}_3$) and Lemma 9 (for the consistency of the coefficients $\hat{\beta}^c_{2\to1,3[0]}$, $\hat{\beta}^c_{2\to1,3[1]}$ and $\hat{\beta}^c_{2\to1,3[3]}$). □

**Remark 13** (Mean estimation: algebraic vs. graphical approach). *In both approaches, the PPCA model is translated into a linear model. However, both estimators in Equations (9) and (57) theoretically differ. The exogeneity assumption and approximation is not made at the same step. In the algebraic approach, the results are first derived without using any approximation. It gives linear models that do not comply with the standard exogeneity assumption. Consequently, an approximation is done at the estimation step since the parameters $\hat{\mathcal{B}}^c_{2\to1,3[0]}$, $\hat{\mathcal{B}}^c_{2\to1,3[1]}$ and $\hat{\mathcal{B}}^c_{2\to1,3[3]}$ are estimated with the standard linear regression coefficients. In the graphical approach, an approximation is made at the first step when a structural equation model is associated with the graphical model by assuming the exogeneity, i.e. $\mathbb{E}[\epsilon_{.2}|Y_{.1},Y_{.3}]=0$. In practice, for both approaches, the same coefficients are naturally computed, i.e. $\hat{\beta}^c_{j\to k,\ell} = \hat{\mathcal{B}}^c_{j\to k,\ell}$, which leads to the same computed estimators for the mean of $Y_{.1}$.*

While only one simplified graphical model between $Y_{.1}$, $Y_{.2}$ and $Y_{.3}$, displayed in the bottom right graph of Figure 1, was required to construct an estimator of the mean of $Y_{.1}$, the variance and covariance estimations rely on Equation (56) and the following one (associating to the bottom left graph of Figure 1),

$$Y_{.3} = \beta_{3\to1,2[0]} + \beta_{3\to1,2[1]}Y_{.1} + \beta_{3\to1,2[2]}Y_{.2} + \epsilon_{.3},\tag{59}$$

assuming $\mathbb{E}[\epsilon_{.3}|Y_{.1},Y_{.2}] = 0$ and where $\beta_{3\to1,2[0]}$, $\beta_{3\to1,2[1]}$ and $\beta_{3\to1,2[2]}$ are the intercept and the coefficients of the linear regression of $Y_{.3}$ on $Y_{.1}$ and $Y_{.2}$.

Using Equations (56) and (59) and Lemmas 9, 10, one can derive some estimators for the variance and the covariances of $Y_1$.

**Proposition 14** (Variance and covariances formulae resulting from the graphical approach when $p = 3$ and $r = 2$). *Under the two equations (56) and (59), assuming **A1.** and also $\beta^c_{3\to1} \neq 0$, $\beta^c_{2\to1,3[1]} \neq 0$ and $\mathrm{Var}(Y_{.3}) \neq 0$, one can construct an estimator of the variance of the MNAR variable $Y_{.1}$ and its covariances as follows*

$$\widehat{\mathrm{Var}}(Y_{.1}) := \frac{\widehat{\mathrm{Var}}(Y_{.3})}{\hat{\beta}^c_{3\to1}} \frac{1}{\hat{\beta}^c_{2\to1,3[1]}} \left( \frac{\widehat{\mathrm{Cov}}(Y_{.2},Y_{.3})}{\widehat{\mathrm{Var}}(Y_{.3})} - \hat{\beta}^c_{2\to1,3[3]} \right),\tag{60}$$

$$\widehat{\mathrm{Cov}}(Y_{.1},Y_{.2}) := \frac{1}{\hat{\beta}^c_{3\to1,2[1]}} \left( \frac{\widehat{\mathrm{Cov}}(Y_{.2},Y_{.3})}{\widehat{\mathrm{Var}}(Y_{.2})} - \hat{\beta}^c_{3\to1,2[2]} \right) \widehat{\mathrm{Var}}(Y_{.2}),\tag{61}$$

$$\widehat{\mathrm{Cov}}(Y_{.1},Y_{.3}) := \frac{1}{\hat{\beta}^c_{2\to1,3[1]}} \left( \frac{\widehat{\mathrm{Cov}}(Y_{.2},Y_{.3})}{\widehat{\mathrm{Var}}(Y_{.3})} - \hat{\beta}^c_{2\to1,3[3]} \right) \widehat{\mathrm{Var}}(Y_{.3}),\tag{62}$$

*where $\hat{\beta}^c_{3\to1,2[1]}$, $\hat{\beta}^c_{3\to1,2[2]}$ and $\hat{\beta}^c_{3\to1}$ are some estimators of $\beta^c_{3\to1,2[1]}$, $\beta^c_{3\to1,2[2]}$ and $\beta^c_{3\to1}$ given in (59).*

*These estimators are consistent under additional Assumption **A4.**.*

*Proof.* To derive some estimators of the variance and covariances of the MNAR variable $Y_{.1}$, one want to obtain the following formulae:

$$\mathrm{Var}(Y_{.1}) = \frac{\mathrm{Var}(Y_{.3})}{\beta^c_{3\to1}} \frac{1}{\beta^c_{2\to1,3[1]}} \left( \frac{\mathrm{Cov}(Y_{.2}, Y_{.3})}{\mathrm{Var}(Y_{.3})} - \beta^c_{2\to1,3[3]} \right), \tag{63}$$

$$\mathrm{Cov}(Y_{.1}, Y_{.2}) = \frac{1}{\beta^c_{3\to1,2[1]}} \left( \frac{\mathrm{Cov}(Y_{.2}, Y_{.3})}{\mathrm{Var}(Y_{.2})} - \beta^c_{3\to1,2[2]} \right) \mathrm{Var}(Y_{.2}), \tag{64}$$

$$\mathrm{Cov}(Y_{.1}, Y_{.3}) = \frac{1}{\beta^c_{2\to1,3[1]}} \left( \frac{\mathrm{Cov}(Y_{.2}, Y_{.3})}{\mathrm{Var}(Y_{.3})} - \beta^c_{2\to1,3[3]} \right) \mathrm{Var}(Y_{.3}). \tag{65}$$

Using Equation (55), one has

$$\mathrm{Cov}(Y_{.1}, Y_{.3}) = \mathrm{Var}(Y_{.1})\beta_{3\to1},$$
$$\mathrm{Cov}(Y_{.3}, Y_{.1}) = \mathrm{Var}(Y_{.3})\beta_{1\to3},$$

so

$$\mathrm{Var}(Y_{.1}) = \frac{\mathrm{Var}(Y_{.3})\beta_{1\to3}}{\beta_{3\to1}}.$$

Considering the graphical model in the bottom left graph of Figure 1,

$$\mathrm{Cov}(Y_{.2}, Y_{.3}) = \beta_{2\to1,3[1]}\beta_{1\to3}\mathrm{Var}(Y_{.3}) + \beta_{2\to1,3[3]}\mathrm{Var}(Y_{.3}) \qquad \text{(by Lemma 10)}$$

$$\Rightarrow \beta_{1\to3} = \frac{1}{\beta_{2\to1,3[1]}} \left( \frac{\mathrm{Cov}(Y_{.2}, Y_{.3})}{\mathrm{Var}(Y_{.3})} - \beta_{2\to1,3[3]} \right)$$

$$\Rightarrow \beta_{1\to3} = \frac{1}{\beta^c_{2\to1,3[1]}} \left( \frac{\mathrm{Cov}(Y_{.2}, Y_{.3})}{\mathrm{Var}(Y_{.3})} - \beta^c_{2\to1,3[3]} \right) \tag{66}$$

where the last implication is given by Lemma 9 and Assumption **A1.**, giving also

$$\beta_{3\to1} = \beta^c_{3\to1},$$

which leads to Equation (63).

By (55), the covariances can be expressed in two different ways,

$$\mathrm{Cov}(Y_{.1}, Y_{.2}) = \beta_{2\to1}\mathrm{Var}(Y_{.1}) \quad \text{and} \quad \mathrm{Cov}(Y_{.1}, Y_{.3}) = \beta_{3\to1}\mathrm{Var}(Y_{.1}), \tag{67}$$
$$\mathrm{Cov}(Y_{.1}, Y_{.2}) = \beta_{1\to2}\mathrm{Var}(Y_{.2}) \quad \text{and} \quad \mathrm{Cov}(Y_{.1}, Y_{.3}) = \beta_{1\to3}\mathrm{Var}(Y_{.3}). \tag{68}$$

In (67), the coefficients $\beta_{2\to1}$ and $\beta_{3\to1}$ can be estimated on the complete case using Lemma 9, but the variance of $Y_{.1}$ has still to be taken care of. Instead of potentially propagate error from (63), we propose to favor the expressions given in (68) to evaluate the covariances.

Focusing on (68), the coefficient $\beta_{1\to3}$ is given in (66) and $\beta_{1\to2}$ can be obtained using the same method, based on the reduced graphical model in the bottom right graph of Figure 1 (by Assumption **A1.**), so that

$$\beta_{1\to2} = \frac{1}{\beta^c_{3\to1,2[1]}} \left( \frac{\mathrm{Cov}(Y_{.2}, Y_{.3})}{\mathrm{Var}(Y_{.2})} - \beta^c_{3\to1,2[2]} \right).$$

Therefore, by plugging it in (68), Equations (64) and (65) are obtained.

The natural estimators for $\mathrm{Var}(Y_{.1})$, $\mathrm{Cov}(Y_{.1}, Y_{.2})$ and $\mathrm{Cov}(Y_{.1}, Y_{.3})$ are then given by (60), (61) and (62). They are consistent given that all the quantites involved are consistent, by using **A4.** (for the consistency of $\widehat{\mathrm{Var}}(Y_{.2})$, $\widehat{\mathrm{Var}}(Y_{.3})$ and $\widehat{\mathrm{Cov}}(Y_{.2}, Y_{.3})$) and Lemma 9 (for the consistency of $\hat{\beta}^c_{j\to k,\ell}$). $\qquad \square$

**Remark 15** (Var-covariance estimation: algebraic vs. graphical approach). *As for the mean, the exogeneity assumption is required in the last step of the algebraic approach to estimate coefficients and in the first step of the graphical approach to obtain structural equation models. However, contrary to the estimator suggested for the mean, the estimators in both graphical and algebraic approaches here differ (compare* (10) *with* (60), (61) *and* (62)). *Indeed, the algebraic approach is based on the use of conditionality, while the graphical one relies on graphical results standing for the linear models when exogeneity holds.*

# G  PPCA with MAR data

The following proposition is an adaptation of our method to handle MAR data, called **MAR** in Section 4.1, inspired by [18, Theorems 1, 2, 3]. In this case, the missing variables are assumed to be MAR indexed by $\mathcal{M}$. We assume the following:

**A1$_{\text{MAR}}$.** $(B_{.j'})_{j' \in \mathcal{J}}$ is invertible.

**A2$_{\text{MAR}}$.** $\forall m \in \mathcal{M}, Y_{.m} \perp\!\!\!\perp \Omega_{.m} | (Y_{.k})_{k \in \overline{\{m\}}}$

**A3$_{\text{MAR}}$.** $\forall m \in \mathcal{M}$, the complete-case coefficients $\mathcal{B}^c_{m \rightarrow \mathcal{J}[0]}$ and $\mathcal{B}^c_{m \rightarrow \mathcal{J}[k]}, k \in \mathcal{J}$ can be consistently estimated.

**A5$_{\text{MAR}}$.** $\forall \ell \in \bar{\mathcal{J}}$, for all set $\mathcal{H} \subset \mathcal{J}_{-j}$ such that $|\mathcal{H}| = r - 1$, $(B_{.\ell} \quad (B_{.j'})_{j' \in \mathcal{H}})$ is invertible,

**A6$_{\text{MAR}}$.** $\forall m \in \mathcal{M}, \forall \ell \in \bar{\mathcal{J}} \backslash \mathcal{M}, \forall j \in \mathcal{J}, \ Y_{.m} \perp\!\!\!\perp \Omega_{.\ell} | (Y_{.k})_{k \in \overline{\{m\}}}.$

**A8$_{\text{MAR}}$.** $\forall m \in \mathcal{M}, \forall \ell \in \overline{\{m\}} \backslash \mathcal{J}$, for all set $\mathcal{H} \subset \mathcal{J}$ such that $|\mathcal{H}| = r - 1$, the complete-case coefficients $\mathcal{B}^c_{m \rightarrow \ell, \mathcal{H}[0]}$ and $\mathcal{B}^c_{m \rightarrow \ell, \mathcal{H}[k]}, k \in \{\ell\} \cup \mathcal{H}$ can be consistently estimated.

**Proposition 16** (Expectation, variance and covariances formulae for a MAR variable when $p = 3$ and $r = 2$). *Consider the PPCA model* (1). *Under Assumptions A1$_{\text{MAR}}$. and A2$_{\text{MAR}}$., one can construct the estimators of the mean, the variance and the covariances with a pivot variable for any MAR variable $Y_{.m}, m \in \mathcal{M}$, as follows*

– *the mean of the missing variable*

$$\hat{\alpha}_m = \hat{\mathcal{B}}^c_{m \rightarrow \mathcal{J}[0]} + \sum_{j \in \mathcal{J}} \hat{\mathcal{B}}^c_{m \rightarrow \mathcal{J}[j]} \hat{\alpha}_j,$$

*with $\mathcal{J}$ the pivot variables set,*

– *the variance of the missing variable*

$$\widehat{\text{Var}}(Y_{.m}) = \hat{Q}^c_{\text{MAR}} + \sum_{j \in \mathcal{J}} (\hat{\mathcal{B}}^c_{m \rightarrow \mathcal{J}[j]})^2 \widehat{\text{Var}}(Y_{.j})$$
$$+ 2 \sum_{(j<k) \in \mathcal{J}} \hat{\mathcal{B}}^c_{m \rightarrow \mathcal{J}[j]} \hat{\mathcal{B}}^c_{m \rightarrow \mathcal{J}[k]} \widehat{\text{Cov}}(Y_{.j}, Y_{.k}),$$

*with*

$$\hat{Q}^c_{\text{MAR}} = \left( \widehat{\text{Var}}(Y_{.m}) | \Omega_{.m} = 1 \right)$$
$$- \left( \widehat{\text{Cov}}((Y_{.j})_{j \in \overline{\{m\}}}, Y_{.m}) \widehat{\text{Var}}((Y_{.j})_{j \in \overline{\{m\}}})^{-1} \widehat{\text{Cov}}((Y_{.j})_{j \in \overline{\{m\}}}, Y_{.m})^T | \Omega_{.m} = 1 \right).$$

– *the covariances between the missing variable and a pivot variable, for all $\ell \in \mathcal{J}$,*

$$\widehat{\text{Cov}}(Y_{.m}, Y_{.\ell}) = \hat{\mathcal{B}}^c_{m \rightarrow \mathcal{J}[0]} \hat{\alpha}_\ell + \hat{\mathcal{B}}^c_{m \rightarrow \mathcal{J}[\ell]} (\widehat{\text{Var}}(Y_{.\ell}) + \hat{\alpha}_\ell^2)$$
$$+ \sum_{k \in \mathcal{J}_{-\ell}} \hat{\mathcal{B}}^c_{m \rightarrow \mathcal{J}[k]} (\widehat{\text{Cov}}(Y_{.\ell}, Y_{.k}) + \hat{\alpha}_\ell \hat{\alpha}_k) - \hat{\alpha}_m \hat{\alpha}_\ell$$

*Under Assumption A3$_{\text{MAR}}$. and A4., these estimators are consistent.*

*In addition, under Assumption A5$_{\text{MAR}}$., A6$_{\text{MAR}}$. and A7., one can construct the estimator of the covariance between a MAR variable $Y_{.m}$ for $m \in \mathcal{M}$ and any not pivot variable as follows*

– *the covariances between the missing variable and any not pivot variable, for all $\ell \in \overline{\{m\}} \backslash \mathcal{J}$, choose $r - 1$ variable indexes in $\mathcal{J}$ to form the set $\mathcal{H} \cup \mathcal{J}$ such that $|\mathcal{H}| = r - 1$*

$$\widehat{\text{Cov}}(Y_{.m}, Y_{.\ell}) = \mathcal{B}^c_{m \rightarrow \ell, \mathcal{H}[0]} \hat{\alpha}_\ell + \hat{\mathcal{B}}^c_{m \rightarrow \ell, \mathcal{H}[\ell]} (\widehat{\text{Var}}(Y_{.\ell}) + \hat{\alpha}_\ell^2)$$
$$+ \sum_{k \in \mathcal{H}} \hat{\mathcal{B}}^c_{m \rightarrow \ell, \mathcal{H}[k]} (\widehat{\text{Cov}}(Y_{.\ell}, Y_{.k}) + \hat{\alpha}_\ell \hat{\alpha}_k) - \hat{\alpha}_m \hat{\alpha}_\ell$$

*Under the additional Assumptions $A8_{MAR}$. and $A9$. this estimator is consistent.*

*Proof.* The proof follows exactly the same direction than in Proposition 4, 5 and 8. The only difference is that the regressions used are not the same.

For the sake of clarity, consider the same toy example as in Section 3.1 where $p = 3, r = 2$, in which only one variable can be missing (at random), and fix $\mathcal{M} = \{1\}$ and $\mathcal{J} = \{2, 3\}$. Note that here the MAR mechanism leads to $\mathbb{P}(\Omega_{.1} = 0|Y_{.1}, Y_{.2}, Y_{.3}) = \mathbb{P}(\Omega_{.1} = 0|Y_{.2}, Y_{.3})$.. The goal is to estimate the mean of $Y_{.1}$, without specifying the distribution of the missing-data mechanism and using only the observed data.

Assumption $A1_{MAR}$. allows to obtain linear link between the MAR variable $Y_{.1}$ and the pivot variables $(Y_{.2}, Y_{.3})$. In particular, one has

$$Y_{.1} = \beta_{1\to2,3[0]} + \beta_{1\to2,3[2]}Y_{.2} + \beta_{1\to2,3[3]}Y_{.3} + \zeta,$$

with $\beta_{1\to2,3[0]}, \beta_{1\to2,3[2]}$ and $\beta_{1\to2,3[3]}$ the intercept and coefficients standing for the effects of $Y_{.1}$ on $Y_{.2}$ and $Y_{.3}$, and with

$$\zeta = -\mathcal{B}_{1\to2,3[2]}\epsilon_{.2} - \mathcal{B}_{1\to2,3[3]}\epsilon_{.3} + \epsilon_{.1}$$

Assumption $A2_{MAR}$., *i.e.* $Y_{.1} \perp\!\!\!\perp \Omega_{.1}|Y_{.2}, Y_{.3}$, is required to obtain identifiable and consistent parameters of the distribution of $Y_{.1}$ given $Y_{.2}, Y_{.3}$ in the complete-case when $\Omega_{.1} = 1$, denoted as $\beta^c_{1\to2,3[0]}$, $\beta^c_{1\to2,3[2]}$ and $\beta^c_{1\to2,3[3]}$,

$$(Y_{.1})_{|\Omega_{.1}=1} = \beta^c_{1\to2,3[0]} + \beta^c_{1\to2,3[2]}Y_{.2} + \beta^c_{1\to2,3[3]}Y_{.3} + \zeta^c,$$

with

$$\zeta^c = -\mathcal{B}^c_{1\to2,3[2]}\epsilon_{.2} - -\mathcal{B}^c_{1\to2,3[3]}\epsilon_{.3} + \epsilon_{.1}$$

(In the MNAR case, the regression of $Y_{.1}$ on $(Y_{.2}, Y_{.3})$ is prohibited, as $A2_{MAR}$. does not hold. That is why we used the regression of $Y_{.2}$ on $Y_{.1}$ and $Y_{.3}$.);

Using again $A2_{MAR}$., one has

$$\mathbb{E}\left[Y_{.1}|Y_{.2}, Y_{.3}, \Omega_{.1} = 1\right] = \mathbb{E}\left[\beta^c_{1\to2,3[0]} + \beta^c_{1\to2,3[2]}Y_{.2} + \beta^c_{1\to2,3[3]}Y_{.3}|Y_{.2}, Y_{.3}\right] + \mathbb{E}[\zeta^c|Y_{.2}, Y_{.3}],$$

and taking the expectation leads to

$$\mathbb{E}\left[Y_{.1}\right] = \beta^c_{1\to2,3[0]} + \beta^c_{1\to2,3[2]}\mathbb{E}\left[Y_{.2}\right] + \beta^c_{1\to2,3[3]}\mathbb{E}\left[Y_{.3}\right],$$

given that $\mathbb{E}[\epsilon_{.k}] = 0, \ \forall k \in \{1, 2, 3\}$.

One obtains

$$\alpha_1 = \beta^c_{1\to2,3[0]} + \beta^c_{1\to2,3[2]}\alpha_2 + \beta^c_{1\to2,3[3]}\alpha_3$$

A natural estimator for $\alpha_1$ is

$$\hat{\alpha}_1 = \hat{\beta}^c_{1\to2,3[0]} + \hat{\beta}^c_{1\to2,3[2]}\hat{\alpha}_2 + \hat{\beta}^c_{1\to2,3[3]}\hat{\alpha}_3,$$

which is consistent using Assumption $A3_{MAR}$. and $A4$.. $\qquad\square$