[Reviews · NeurIPS 2020]

Review 1

Summary and Contributions: The paper develops a missing data imputation method when the data follows a low-rank assumption and is missing not-at-random. Previous approaches to this problem models the missing mechanism (e.g. using a logistic regression) and are suitable for low-rank fixed effects data. The paper's approach applies to a slightly broader class of models (probabilistic PCA) and uses the idea of several surrogate regressions using "pivot variables" (those that are not affected by the missing-not-at-random metchanism) to construct mean and variance estimates for the missing variables.

Strengths: Soundness: Good. The toy example of Sec3.1 helps ground the solution strategy, and the graphical arguments give good intuition for why the strategy can work. The synthetic and semi-synthetic experiments illustrate some of the benefits of the approach. Significance: Good. Most real-world datasets have missing values in them, and several low-rank models have been used fruitfully on these datasets. The proposed approach may offer a more flexible approach in using such datasets.

Weaknesses: Clarity: Room for improvement. Beyond the toy example of Sec 3.1, it may help to take a real world example, e.g. user volunteered ratings for movies (which exhibit missing not-at-random effects since users are more likely to rate the movies they like). Establish what the PPCA model implies for this data, and how it is a more flexible fit than, say, logistic regression to estimate missingness as in a fixed-effects low rank model. Also clarify what assumptions A1, A2 (and later, A3, A4) require for this dataset. Without such examples, it is essentially impossible to verify if A3, A4 are reasonable or completely unreasonable assumptions, and whether we can ever hope to test them in realistic scenarios. Experiments: The main text will benefit from bringing an additional experiment from the supplementary. Suppose the data follows the fixed-effects low rank model; how does Algorithm 1 compare against MNARparam? In other words, what is the price we pay for flexibly reasoning about a PPCA model when the data is generated by a simpler model? (Usually, this manifests as needing more data to draw reliable imputations).

Correctness: Yes. However, I have not stepped through the extensive supplementary material -- only checked that the proof strategy seems reasonable. There are two vulnerabilities: (1) if some of the true coefficients are 0 (e.g. in Eqn8, if B^c_{2->1,3[1]} is 0, then the solution strategy will attempt to divide by 0. I didn't see an immediate proof that for at least one of the pivot variables, this coefficient is guaranteed to be non-zero. (2) Going from B^c coefficients (which depend on the unknown loading matrix B) to estimated regression coefficients hat{B}^c. It's unclear whether the estimated coefficients, e.g. by OLS, should be any good approximation of the true coefficients B^c.

Clarity: Clarity can be improved. Please see comments throughout the review for ways to address this, via (1) illustration, (2) additional discussion, (3) additional experiment showing trade-off against a prior approach.

Relation to Prior Work: Adequate. The Broader Impacts section currently only highlights positive impacts of the work, e.g. in genome studies, collaborative filtering etc. These missing data imputation techniques may also be used in more dubious circumstances -- e.g. when individuals choose to opt out of volunteering their social media activity due to privacy concerns, these techniques can impute their preferences using "similar" users (via the low rank assumption). Though this technique may neither enable or hinder such uses, complementary research in privacy-preserving collaborative filtering and other applications may be needed.

Reproducibility: Yes

Additional Feedback: I have read the other reviews and the authors' feedback. With the addition of the recommender system experiment, walking the readers through how (1) the MNAR and PPCA model apply in this setting, (2) selecting the hyper-parameters for the imputation algorithm, (3) showing how the imputations compare with prior algorithms, helps make a strong case for the proposed method. If the authors re-arrange the paper to improve clarity (as the reviews point out, and as they promise in their feedback), the paper can be substantially stronger. There are a few lingering questions from the reviews that the authors should address in the paper at a minimum -- (1) a discussion on a stage-wise approach to imputation (and why that may not be necessary for their sequence of regressions), (2) given that some of the linear coefficients can be zero, what must a practitioner do when one of the regressions estimate a coefficient close to 0 that is then used in the denominator of other estimates. ---------- Section 2 will benefit from an illustration of a matrix, with the M set and pivot variables highlighted. Even better if the illustration is grounded in an example like movie item ratings. Re: Eqn8, one could derive analogous equation by considering the effects of Y.3 on Y.2 and Y.1 Is it straightforward to combine the resulting mean estimates of Y.1 by combining these 2 equations? Is there a statistically appealing way to combine them? There is a brief comment after Eqn11, but this point may benefit from further discussion. Algorithm 1 requires knowing r and sigma. Is there a technique (e.g. cross-validation) that can be used to set them? Does that technique require MAR/MCAR/fully observed data? The experiments will be more realistic with empirically tuned r,sigma settings (rather than oracle values). Leaving this entirely to future work (without at least an initial heuristic approach) leaves Algorithm1 still incomplete for practical uses -- Alg1 seems robust to mis-specified r's in the supplementary experiments, so even a reasonable heuristic may already be enough. Expt setting: throughout the text, it seems that the no. of pivot variables is equal to the rank r. In the experiments however, the pivot variables are greater than r. Can this be clarified please? I thought that if pivot variables > r, then all r-subsets are taken and Alg1 is run for each of them, and the median estimate is returned. In line 252, it will be useful to clarify this. Please make the code available broadly as an R package online. Typos: 41: litterature -> literature 44: whereas -> when (multiple places throughout the text) 49: "as that of considering simple models" -> unclear, please rephrase this.


Review 2

Summary and Contributions: The authors propose a method for parameter estimation and missing data imputation in the probabilistic PCA model when data are missing not at random (MNAR). Unlike previous work, their identifiability results apply to any monotone missingness mechanism. Simulations and application indicate performance which is superior to existing missing data methods.

Strengths: I think this is an important problem and the empirical results are quite compelling. The authors are very thorough and include an extensive Appendix with detailed results.

Weaknesses: The theoretical exposition is dense, the notation is difficult and the theory is not always well described. In particular, I had to work hard to parse the key assumptions and how the results relate to existing work. First, I wonder why "Model Identifiability" and "Proposition 1" are not stated more formally as a Theorem with key assumptions and the result. One assumption that I think the authors make (though I'm not sure) is that the functional form of F_m is known. This assumption is made in Miao et al (2016) when identifiability is used. Assumption A.1 is also crucial but took me some time to parse. How strong of an assumption is this? What is meant by "any variable is generated by all latent variables"? In the proof in Section A.1 if Sigma_{12}=0 then you aren't guaranteed identifiability? Is Sigma_{12} != 0 an assumption implied by assumption A.1? I think the authors need to be much more upfront about this assumption: clearly if B is too sparse Assumption A.1 will be violated and there will be no identifiability. I'd like to see a bit more discussion about how this relates to Miao et al, since the authors explicitly contrast their own results with those in this paper. In particular, what assumptions guarantee identifiability in your setting but not theirs? They provide results on identification with fully observed covariates. Since pivot variables can be fully observed, is there a comparison between these results? UPDATE: The authors addressed many of my concerns. While the paper introduces some important results, I think the biggest issue that I have still remains: the clarity of the paper is still a major issue for me, and it's not clear how much this will be re-arranged conditional on an acceptance. As such I think this paper is still a "weak accept".

Correctness: I believe so, but some more clarification on some of the above points would help.

Clarity: The clarity is one of the major criticisms I have of this work. There are several awkward phrases, misspellings and other places where incorrect English is used. On top of this, I found some of the notation, especially in the "algebraic arguments" Section to be very unclear. I don't know if this is a lack of familiarity of the relevant literature on my part, or whether this is nonstandard, but I had a hard time parsing what was going on here. What is the conclusion of the two "using ... arguments" sections? I couldn't identify what was being argued!

Relation to Prior Work: Yes, the authors to a nice job extensively citing relevant literature. As noted above, I would like to see a bit more discussion about how the results relate to Miao et al (2016).

Reproducibility: Yes

Additional Feedback:


Review 3

Summary and Contributions: This paper addresses an important problem when it comes to modeling with MNAR missingness. It shows how to learn an unbiased PPCA with MNAR missing data and furthermore shows that the resulting model is identifiable under certain conditions. --- post-rebuttal: I appreciate the effort that the authors managed to put together an extra experiment to address many of my previous questions. I agree with the rest of the reviewers that this paper provides a valuable theoretical contribution to modeling MNAR data. Therefore, I raise my rating to 6. However, the clarity of the paper still needs to be improved and there are also some important technical concerns raised by reviewer #1 that need to be addressed.

Strengths: Most of the existing works on learning with missing data assume MAR or MCAR, which in reality might not be true. This work proposes an algorithm to learn a PPCA under the most general MNAR assumption and also show that this model can produce unbiased imputation. In addition, it also discusses the identifiability of PPCA, which is of great theoretical interest especially under the challenging MNAR setting.

Weaknesses: The proposed model is somewhat restricted: it relies on the known set of MNAR variables and pivot variables, which usually are hard to identify in many real-world problems. The requirements for identifiability in Proposition 1 are also hard to satisfy in practice including self-masking mechanism, known noise level, strictly monotone F functions, etc.

Correctness: I didn't check the proofs in the supplemental materials in detail. Please also see the comments in the additional feedback section.

Clarity: I feel this paper is a bit hard to follow as justifications of most core results are deferred to the supplemental materials; it'd be great if a sketch of proof is provided in the main paper. Some of the notations are used before they are properly defined, and it somewhat hurts readability. Please see the comments in the additional feedback section.

Relation to Prior Work: Sufficient pointers to related work are provided and summarized.

Reproducibility: Yes

Additional Feedback: I want to make it clear up front that I'm not familiar with this line of research on the justification of model identifiability and I didn't go through the supplemental materials in detail. My comments below are merely providing the authors with some idea what concerns or misunderstandings might come from the general audience in the hope that the authors can improve their paper and make it more inviting. In line 92, the parameters (\alpha, \Sigma) are mentioned, but it seems that \Sigma has been defined before, at least not in Equation 1. It's better to explain what the notation \mathcal{B} with subscript in the form of a->b,c[d] means right before the first time it is used in Equation 7. In Section 3.1, under Equation 8 it says that the regression of Y_{.1} on (Y_{.2}, Y_{.3}) is prohibited as A.2 does not hold, and the same statement is made in the graphical model paragraph. It seems to me that if we take the graphical model in Figure 1 for example and make Y_{.1} depends on Y_{.2} and Y_{.3}, \Omega_{.1} is independent of Y_{.2} given Y_{.1} and Y_{.3} as it is blocked by Y_{.1}. It's still true for the general case that follows A.2 as \overline{\{j\}} always includes the entire \mathcal{M} that the mask \Omega_{.m} for m \in \mathcal{M} associated with MNAR variables depends on. Did I miss anything? In Section 3.2, the index m used in Lemma 2 and Definition 3 is not defined. Probably m \in \mathcal{M}? What if |\mathcal{M}| > 1? For step 2 in Algorithm 1 on computing those \alpha's, I was under the impression that it is computed using only those data cases with all of the involved entries in Y observed. If I'm not mistaken, I'm concerned about the data efficiency of the first part of Algorithm 1 (the left side of Algorithm 1). Also for Algorithm 1, have the authors considered iteratively improving the stagewise algorithm? Specifically, once the missing variables are imputed at the end, is it possible to make use of those imputations to further refine the estimation of the mean, variance, and covariance in the first step and iterate until convergence? All the experiments in the main paper evaluate how biased different models are. As model identifiability is one of the main contributions of the paper, I'm hoping to see some empirical assessment on this, for example, to demonstrate that the proposed method actually learned an identifiable model. What is the formal definition of "prediction error" used in the experiments in Section 4.1? For the experimental results in Section 4.1, error bars are shown in Figure 2 and Figure 3. However, there is no mention of where the stochasticity comes from--whether the results come from learning with multiple data matrices and missing values that are generated according to the same setting or repeatedly running the same algorithm with different random seeds. For the experiment in Section 4.2, it'd be great if the authors could provide details about the TraumaBase dataset. The paper mentioned that HR.ph has an initial missing rate of 1%; does this imply that this data contains missing values originally? What does the missing rate look like for each variable? Artificially injecting a great amount of missingness (50%) in a controlled way (a logistic self-masked mechanism) could dilute the actual missing mechanism that comes along and make the problem easier. It's understandable that the authors did this so that they can evaluate the imputation performance. Nevertheless, the authors could also consider assessing their model on supervised learning performance if any fully-observed variable could be made a labeled target. Another concern of the experiment with clinical data is that physiological variables might have a quite complicated generative process that cannot be properly modeled by linear models like PPCA. Given the potential model mismatch in this sense, I wonder whether the modeling overhead of addressing the more complicated MNAR missingness would jeopardize the ability to learn a useful model for downstream tasks. Recent work on the somewhat related topic--identifiability of deep generative models--which could possibly be extended to the missing data setting: Khemakhem, Ilyes, et al. "Variational autoencoders and nonlinear ICA: A unifying framework." International Conference on Artificial Intelligence and Statistics. 2020. Roeder, Geoffrey, Luke Metz, and Diederik P. Kingma. "On Linear Identifiability of Learned Representations." arXiv preprint arXiv:2007.00810 (2020). minor grammatical errors: line 51: capitalization line 176: repeated commas


Review 4

Summary and Contributions: In paper proposes a theoretical approach to estimation and imputation of missing data in the case of MNAR scenario, which is one of the most difficult cases in missing data. The unavailability of the data in this case depends on both observed and unobserved data. The result of the presented theory is an algorithm for imputating missing values.

Strengths: The idea of the paper is interesting and makes a theoricaly contribution. The theory is rather correct but I have concerns about experiments which I discuss below in detail.

Weaknesses: Authors made numerical experiments on synthetic data which confirm good estimations of mean and covariance of the missing variables. I have concerns since only one real dataset was used. The authors should use more real data sets and compare to other methods e.g. https://arxiv.org/pdf/1808.01684.pdf, which imputates missing values using deep generative models for MNAR strategy, https://arxiv.org/pdf/1705.02737.pdf - multiple Imputation using denoising auto encoders (obtained much better results than MICE) to confirm the effectiveness of your method.

Correctness: I have discussed most of my concerns above.

Clarity: Yes

Relation to Prior Work: Yes

Reproducibility: Yes

Additional Feedback: Post rebuttal ============================================== I do not hold a strong opinion and still consider it as a borderline paper after reading authors' responses. I agree that the clarify of the paper needs to be improved and the some technical issues need to be addressed. I would like to maintain my original score 6.

[Author Response · NeurIPS 2020]

We would like to thank the reviewers for their thorough reading of the article and their many pertinent remarks, which help to improve the clarity. In the final version, we will address all comments on form. To address the reviewers' main concerns and better show the extent and feasibility of our methodology, we respond by adding an application on a recommendation system data. We consider the Jester dataset [2] of 5000 users who rated jokes, with 27% of missing values. The low-rank assumption for the loading matrix (allowed by Assump. 1) makes sense: any variable (i.e. user preferences) can be expressed as a linear combination of $r$ latent variables[1] (hence, a "fully connected PPCA"). The first latent variable opposes individuals who like jokes about physics but dislike jokes about sexuality, and conversely.

**MNAR mechanism. (R1, R3)**   Considering MNAR and self-masking values is plausible because users only rate jokes they like or dislike strongly or might be ashamed to assume their taste for sexual jokes. Note that the **self-masked assumption** is required only for the identifiability but the estimation strategy is also derived for **general MNAR mechanisms** (allowed by Assump. A2) where the missingness may depend on other missing variables. Assump. A2 means that a user's non-response for the sexual joke given all jokes may depend on the scores of the sexual and physical jokes but not on the musical and computer jokes.

**Selecting the number $r$ of latent variables and estimating the noise variance. (R1)**   To select $r$, one could use complete observations only but this is not possible when the number of features is large. As an alternative, we used both a cross-validation strategy assuming M(C)AR mechanism as detailed in [3] and also a beta implementation (that we coded) of a CV assuming MNAR mechanism. The second one is dependent on the chosen mechanism. As noted by the reviewers, Algorithm 1 is robust to a misspecification of the rank and thus a reasonable heuristic may already be enough. Both approaches estimate $r = 5$. CV was also used for Traumabase where oracle values were only used for synthetic data. With $r$ at hand, the noise variance is obtained directly using weighted residual sum of squares as in [3].

**Selecting the $r$ pivot variables. (R1,R3)**   The next step consists in selecting $r$ (M(C)AR) pivot variables (observed or M(C)AR variables imply Assump. A4) on which regressions[2] are performed [3]. Here, because we do not have further information on the missing mechanisms, we select the variables with the lowest missing rate. In Traumabase, the selection was discussed with experts (doctors) who identified M(C)AR variables. To reduce the error committed by a wrong selection of pivot variables, we suggest selecting a bigger set ($> 5$) and computing the final estimator with the median of the estimators over all possible combinations. In Fig. 2, by discarding outliers, this **aggregation approach** is more robust than selecting only $r$ pivot variables.

**Additional experiments. (R1,R4)**   Then, we test our method by introducing additional MNAR values on one variable (containing 33% NA) using a self-masked mechanism leading to 65% NA. In Fig. 1, **our method (`MNAR`) outperforms all the others on rating data** including `Deep` [1] which imputes MNAR values using deep generative models (R4)[4]. The parametric method `MNARparam` is not displayed as it does not scale on such large data. The code for the whole methodology was already available, but now recast as a **beta version of package** (R1) and submitted soon on CRAN.

Fig. 1: Prediction error (difference between true values and predicted ones) for the Jester dataset, the mean imputation corresponding to an error of 1. The process of drawing additional MNAR values and predicting them is repeated 10 times which gives the stochasticity.

Fig. 2: Synthetic data from Section 4.1, with Algorithm 1 performed with aggregation (`MNARagg`) or not (`MNARnoagg`). True values in red, estimated values (means, variance, cov) in boxplot. For a given set of PPCA parameter, the stochasticity comes from the process of drawing 20 times the latent variables, the additive noise and the missing-data pattern (R3).

**Comparison with Miao et al. [4] (R2,R3)**   For one variable $Y \sim \mathcal{N}(\mu, \sigma^2)$, Miao et al. prove identifiability of the variance and the absolute value of the mean, assuming a self-masked mechanism with a known strictly monotone form (including classical Probit and Logit). They cannot get identifiability for the mean (not the absolute value) with Logit. We have used their result to prove variance identifiability in PPCA and provide a genuine proof for the mean without discarding Logit. Secondly, for a specific setting of an heteroscedastic regression model with missing values only in $Y$, where the variance of $Y$ given the observed covariates is injective, they provide identifiability results for the conditional distribution with general MNAR. This setting and the proof are too restricted to be considered in PPCA.

**Supervised learning task on Traumabase. (R3)**   To predict the administration or not of the tranexomic acid (binary variable), we impute explanatory variables before proceeding to the classification task. In Tab. 1, our method gives the smallest prediction error.

| | |
|---|---|
| MNAR | 5.06% |
| EMMAR | 5.82% |
| SoftMAR | 5.45% |
| MNARparam | 5.39% |
| Mean | 5.27% |

Tab. 1: Mean of prediction error over 10 repetitions.

[1] L. Gondara and K. Wang. Mida: Multiple imputation using denoising autoencoders. In *PAKDD*, 2018.
[2] M. Hahsler. recommenderlab: A framework for developing and testing recommendation algorithms. Technical report, 2015.
[3] J. Josse and F. Husson. Selecting the number of components in principal component analysis using cross-validation approximations. *CS&DA*, 2012.
[4] Wang Miao, Peng Ding, and Zhi Geng. Identifiability of normal and normal mixture models with nonignorable missing data. *JASA*, 2016.

[1] It does not require that the linear combination coefficients are non zero.   [2] assumed to be consistent by Assump. A3, which holds as the noise tends to 0. (R1)   [3] Note that our method is not based on the complete-case of the dataset but on the complete-case of the $r$ ($\ll p$) pivot variables (R3).   [4] Note that this method requires to be trained on a complete dataset.

[Meta-Review · NeurIPS 2020]

There was no consensus among the referees for strongly supporting acceptance and it was felt that the paper was at-best marginal. We hope the reviews would be helpful in developing a stronger version of the paper.